# Regret Bounds for Adversarial Contextual Bandits with General Function Approximation and Delayed Feedback

**Orin Levy**[*]
Blavatnik School of Computer Science
Tel Aviv University
orinlevy@mail.tau.ac.il

**Liad Erez**[*]
Blavatnik School of Computer Science
Tel Aviv University
liaderez@mail.tau.ac.il

**Alon Cohen**
School of Electrical Engineering
Tel Aviv University
and Google Research
alonco@tauex.tau.ac.il

**Yishay Mansour**
Blavatnik School of Computer Science
Tel Aviv University
and Google Research
mansour.yishay@gmail.com

## Abstract

We present regret minimization algorithms for the contextual multi-armed bandit (CMAB) problem over $K$ actions in the presence of delayed feedback, a scenario where loss observations arrive with delays chosen by an adversary. As a preliminary result, assuming direct access to a finite policy class $\Pi$ we establish an optimal expected regret bound of $O(\sqrt{KT \log |\Pi|} + \sqrt{D \log |\Pi|})$ where $D$ is the sum of delays. For our main contribution, we study the general function approximation setting over a (possibly infinite) contextual loss function class $\mathcal{F}$ with access to an online least-square regression oracle $\mathcal{O}$ over $\mathcal{F}$. In this setting, we achieve an expected regret bound of $O(\sqrt{KT \mathcal{R}_T(\mathcal{O})} + \sqrt{d_{\max} D \beta})$ assuming FIFO order, where $d_{\max}$ is the maximal delay, $\mathcal{R}_T(\mathcal{O})$ is an upper bound on the oracle's regret and $\beta$ is a stability parameter associated with the oracle. We complement this general result by presenting a novel stability analysis of a Hedge-based version of Vovk's aggregating forecaster as an oracle implementation for least-square regression over a finite function class $\mathcal{F}$ and show that its stability parameter $\beta$ is bounded by $\log |\mathcal{F}|$, resulting in an expected regret bound of $O(\sqrt{KT \log |\mathcal{F}|} + \sqrt{d_{\max} D \log |\mathcal{F}|})$ which is a $\sqrt{d_{\max}}$ factor away from the lower bound of $\Omega(\sqrt{KT \log |\mathcal{F}|} + \sqrt{D \log |\mathcal{F}|})$ that we also present.

## 1 Introduction

Contextual Multi-Armed Bandit (CMAB) is a natural extension of the well-studied Multi-Armed Bandit (MAB) model that has gained considerable attention over the past decade. (See, e.g., [24, 34]). CMAB describes a sequential decision-making problem with exterior factors that affect the decision taken by the learner. We refer to this side information as the *context*. In this setting, the context $x$ is revealed to the learner at the start of each round. Then, the learner chooses an action $a$ out of a finite set $\mathcal{A}$ containing $K$ actions and suffers a loss for that choice, where the context determines the loss. The learner's goal is to minimize the cumulative loss incurred throughout an interaction of $T$ rounds. In this model, action selection strategies are context-dependent and referred to as *policies*, and the

---

[*] Equal contribution.

39th Conference on Neural Information Processing Systems (NeurIPS 2025).

learner ultimately aims to minimize *regret*, that is, the learner's cumulative loss in comparison to that of the best contextual action selection rule, i.e., the *optimal policy*.

CMAB can describe various real-life online scenarios where there are external factors that affect the loss incurred by any choice of action. One such application is online advertising, where the reaction of a user to a presented advertisement (i.e., clicking or ignoring) is heavily dependent on the user's needs (e.g., if they would like to buy a new car), hobbies, and personal preferences. All of the above can be encoded in the user's browsing history and cookies. Thus, the user's cookies can refer to the external factors that affect the user's implied loss. CMAB has been studied under various assumptions and frameworks, which we review in the sequel. In this paper, we consider the *adversarial CMAB* model (see, e.g., [6, 13]), where the context in each round is chosen by a possibly adaptive adversary from a (possibly infinite) context space $\mathcal{X}$.

Returning to the online advertising example, in such an application, delayed feedback is practically unavoidable. Consider the scenario where a sequence of users enters the application one after another. The algorithm then needs to present them with advertisements, even though the feedback of previous users has not arrived yet. As the application takes time to process each user's feedback, observations will arrive with an inherent delay. In other real-life scenarios such as communication on a physical network, it is also natural to assume that observations arrive at the order in which they are distributed into the network; that is, they arrive in a First-In-First-Out (FIFO) order. In the main theoretical framework of general function approximation that we consider in this work, this additional assumption enables us to obtain highly nontrivial regret guarantees. Such real-life applications motivate the setting of *MAB with delayed feedback*, which has also vastly studied in recent years, either when the environment is adversarial [8, 10, 35] or stochastic [17, 21, 37]. This leads us to the following fundamental question: *What are the achievable regret guarantees in adversarial CMAB under adversarial delayed feedback?* In this work, we address this question by considering the two main settings studied in adversarial CMAB literature, and derive delay-robust algorithms for them.

We start with the simpler setting of *policy class learning*, considered in [5, 12] where the context is stochastic, and in [6] for adversarial contexts. In this setting, the learner has direct access to a finite class $\Pi \subseteq \mathcal{A}^{\mathcal{X}}$ of deterministic mappings from contexts to actions (i.e., policy class), and its performance is compared against the best policy in $\Pi$. We note that using policy class-based approaches, a running-time complexity of $O(|\Pi|)$ is unavoidable in general. It is thus natural to also consider the more challenging setting of *general function approximation* [4, 13, 16, 33], with the goal of obtaining an algorithm that is both computationally efficient and enjoys rate-optimal regret.

In the function approximation framework, the learner has indirect access to a class of loss functions $\mathcal{F} \subseteq [0, 1]^{\mathcal{X} \times \mathcal{A}}$ where each function defines a mapping from context and action to a loss value in $[0, 1]$. They also assume realizability, meaning the true loss function $f_\star$ is within the class, i.e., $f_\star \in \mathcal{F}$. The learner accesses the function class via an online regression oracle, and measures its performance with respect to that of the best contextual policy $\pi_\star : \mathcal{X} \to \mathcal{A}$ of the true CMAB. In this setting, in addition to the standard least-squares regret assumption required from the oracle, our approach will require a stability assumption that will be discussed later. Furthermore, we present novel stability analysis of a Hedge-based version of Vovk's aggregating forecaster [39] and derive an expected regret bound for this setting, assuming finite and realizable loss function classes.

**Summary of our main contributions.** We present delay-adapted algorithms for CMAB with general function approximation and analyze the regret of the proposed methods. In more detail, our main results are summarized as follows:

(1) For the policy class learning setup, we establish a regret bound of $O(\sqrt{KT \log |\Pi|} + \sqrt{D \log |\Pi|})$ where $D$ is the sum of delays. This bound is optimal, as stated in our lower bound in Corollary C.3.

(2) Given access to a finite contextual loss function class $\mathcal{F}$ via an online least-square regression oracle $\mathcal{O}_{\text{sq}}^{\mathcal{F}}$ over $\mathcal{F}$, we present a delay-adapted version of function approximation methods for CMAB, as specified in Algorithm 1. This algorithm can be seen as a delay-adapted version of SquareCB [13], formalized using techniques presented in [15, 26]. For this algorithm we prove in Theorem 4.6 an expected regret bound of $O(\sqrt{KT\mathcal{R}_T(\mathcal{O}_{\text{sq}}^{\mathcal{F}})} + \sqrt{d_{\max}D\beta})$ assuming observations arrive in FIFO order, where $d_{\max}$ is the maximal delay, $\mathcal{R}_T(\mathcal{O})$ is an upper bound on the oracle's regret and $\beta$ is a parameter given by an additional assumption that the oracle in use is sufficiently stable. We also prove (in Appendix C.1) a lower bound showing that without an additional assumption on the oracle, no algorithm can guarantee sublinear regret in the presence of delays in the function approximation

setting. To our knowledge, our work is the first to consider delayed feedback in adversarial CMAB in the fully general function approximation framework.

(3) To complement and strengthen this result, we analyze a hedge-based version of Vovk's aggregating forecaster [39] as an online least-squares regression oracle for finite loss function classes. We show that it enjoys a constant expected regret while also exhibiting nontrivial cumulative stability guarantees for realizable finite classes $\mathcal{F}$ (Theorem 4.11), which implies a constant bound on its stability parameter $\beta \leq \log |\mathcal{F}|$ and in turn an expected regret bound of $O(\sqrt{KT \log |\mathcal{F}|} + \sqrt{d_{\max} D \log |\mathcal{F}|})$ for Algorithm 1 when used with this oracle. We emphasize that the proof of this oracle's stability properties constitutes a significant part of the technical novelties of our work.

## 1.1 Additional related work

**Contextual MAB.** CMAB has been vastly studied over the years, under diverse assumptions, regarding the contexts, the function class or the oracles in use, if any. Previous works divide into two main lines. The first is policy class learning, starting from the fundamental EXP4 algorithm for adversarially chosen contexts [6], to Agarwal et al. [5], Dudik et al. [12] that consider stochastic contexts and present computationally efficient algorithms for this problem. They obtain an optimal regret of $\widetilde{O}(\sqrt{KT \log|\Pi|})$. Dudik et al. [12] also considered constant delayed feedback $d$ and obtained regret bound of $\widetilde{O}(\sqrt{K \log|\Pi|}(d + \sqrt{T}))$.

The second line is the realizable function approximation setting, which has also been studied for stochastic CMAB, starting from Langford and Zhang [23] to Agarwal et al. [3], Simchi-Levi and Xu [33], Xu and Zeevi [40] in which an optimal regret of $\widetilde{O}(\sqrt{TK \log|\mathcal{F}|})$ has been shown, where $\mathcal{F} \subseteq [0,1]^{\mathcal{X} \times \mathcal{A}}$ is a finite contextual reward or loss function class, accessed via an offline regression oracle. Adversarial CMAB has also gained much attention recently, in the following significant line of works [13, 14, 16, 42], where an online regression is being used to access the function class $\mathcal{F}$, with an optimal regret bound of $\widetilde{O}(\sqrt{KT\mathcal{R}_T(\mathcal{O})})$, where $\mathcal{R}_T(\mathcal{O})$ is the oracle's regret.

Regret guarantees for linear CMAB first studied by Abe and Long [2] and the SOTA algorithms are those of Abbasi-Yadkori et al. [1], Chu et al. [11]. Contextual MDPs (which are an extension of MAB, that has multiple states and dynamics) have been studied under function approximation assumptions for both stochastic context [25, 27, 32] and adversarial contexts with Levy et al. [26] being the most relevant to our setting as it studies adversarial CMDP, and inspired our algorithm and analysis. Generalized Linear CMDPs and smooth CMDPs have also been studied, see, e.g., [31, 30].

**Online Learning with Delayed Bandit Feedback.** Delayed feedback has been an area of considerable interest in various online MAB problems in the past few years, with the first work on adversarial MAB with a constant delay $d$ by Cesa-Bianchi et al. [10]. Subsequent results for adversarial MAB with arbitrary delays have been established by [8, 35], with Thune et al. [35] being the first work to introduce the *skipping* technique which adapts to delay sequences that may contain a relatively small number of very large delays. Zimmert and Seldin [43] proposed the first algorithm for adversarial MAB with arbitrary delays that does not require any prior knowledge of the delays.

The study of delayed feedback in MAB has also been extended in several works to more general learning settings. Such settings include linear bandits [19, 37], generalized linear bandits [18], combinatorial semi-bandits [36] and bandit convex optimization [28]. Another prevalent generalization of MAB, in which delayed feedback has been studied, is Reinforcement Learning, specifically tabular MDPs [20, 22, 36], with Jin et al. [20] who first suggested the use of biased delay-adapted loss estimators which inspired our loss estimators used in Algorithm 3.

In CMAB, delayed feedback is far less explored. In the framework of function approximation, Vernade et al. [38] consider the linear case with stochastic delays, and Zhou et al. [41] study generalized linear CMAB with stochastic delays and contexts; both of which are special cases of the general function approximation setting studied in this paper. For stochastic contexts, we believe that obtaining delay-adapted regret bounds in the general function approximation setting can be done by extending the approach of Simchi-Levi and Xu [33], which operates in phases of exponentially increasing lengths. It seems that the presence of delays in this setting will only affect the regret for rounds in which the delay is larger than current batch size, which quickly becomes much larger than the maximal delay. We thus focus on the adversarial setting where it is much less clear how to handle delayed feedback.

## 2 Problem Setup

We consider *adversarial contextual MAB (CMAB) with adversarial delayed bandit feedback*.

**Contextual MAB.** Formally, *CMAB* is defined by a tuple $(\mathcal{X}, \mathcal{A}, \ell)$ where $\mathcal{X}$ is the context space, which is assumed to be large or even infinite, and $\mathcal{A} = \{1, 2, \ldots, K\}$ is a finite action space. $\ell : \mathcal{X} \times \mathcal{A} \to [0, 1]$ forms an expected loss function, that is, for $(x, a) \in \mathcal{X} \times \mathcal{A}$, $\ell(x, a) = \mathbb{E}[L(x, a) \mid x, a]$ where $L(x, a) \in [0, 1]$ is sampled independently from an unknown distribution, related to the context $x$ and the action $a$. In *adversarial CMAB*, the learner faces a sequential decision-making game that is played for $T$ rounds according to the following protocol, for $t = 1, 2, \ldots, T$:
(1) Adversary reveals a context $x_t \in \mathcal{X}$ to the learner; (2) Learner chooses action $a_t \in \mathcal{A}$ and suffers loss $L(x_t, a_t)$.
A *policy* $\pi$ defines a mapping from context to a distribution over actions, i.e., $\pi : \mathcal{X} \to \Delta(\mathcal{A})$. The learner's cumulative performance is compared to that of the best (deterministic) *policy* $\pi_\star : \mathcal{X} \to \mathcal{A}$.

**Delayed feedback.** The learner observes delayed bandit feedback, where the sequence of delays can be adversarial. Formally, delays are determined by a sequence of numbers $d_1, \ldots, d_T \in \{0, 1, \ldots, T\}$. In each round $t$, after choosing an action $a_t$, the learner observes the pairs $(s, L(x_s, a_s))$ for all rounds $s \le t$ with $s + d_s = t$; crucially, only the loss values are delayed, whereas the contexts $x_t$ are each observed at the start of round $t$. We consider a setting where the sequence of delays $(d_t)_{t=1}^T$ as well as the contexts $(x_t)_{t=1}^T$ are generated by an adversary.[0] We denote the sum of delays by $D = \sum_{t=1}^d d_t$ and the maximal delay by $d_{\max} = \max_{t \in [T]} d_t$.

**Learning objective.** We aim to minimize *regret*, which is the difference between the cumulative loss of the learner and that of the best-fixed policy $\pi_\star$, i.e., $\mathcal{R}_T := \sum_{t=1}^T \ell(x_t, a_t) - \ell(x_t, \pi_\star(x_t))$.

We consider two different learning settings for CMAB with delayed feedback. The first is *Policy Class Learning*, in which the CMAB algorithm has direct access to a *finite policy class* $\Pi \subseteq \mathcal{A}^{\mathcal{X}}$. In this setting, the benchmark $\pi_\star$ is the best policy among the class, i.e., $\pi_\star \in \arg\min_{\pi \in \Pi} \sum_{t=1}^T \ell(x_t, \pi(x_t))$. Next, we consider the setting of *Online Function Approximation*, where the CMAB algorithm has access to a *realizable contextual loss* class $\mathcal{F} \subseteq \mathcal{X} \times \mathcal{A} \to [0, 1]$, where realizability means that there exists a function $f_\star \in \mathcal{F}$ such that for all $(x, a) \in \mathcal{X} \times \mathcal{A}$ it holds that $f_\star(x, a) = \ell(x, a)$. Then, the learner's goal is to compete against $\pi_\star(x) \in \arg\min_{a \in \mathcal{A}} f_\star(x, a)$, for all $x \in \mathcal{X}$.

## 3 Warmup: Policy Class Learning

We begin with a simple formulation of the CMAB problem which considers a finite but structureless policy class $\Pi \subseteq \mathcal{A}^{\mathcal{X}}$, indexed by $\Pi = \{\pi_1, \ldots, \pi_N\}$. We remark that in this formulation, the loss vectors $(L(x_t, \cdot))_{t=1}^T$ may also be generated by an adversary.

### 3.1 Algorithm: EXP4 with Delay-Adapted Loss Estimators

For this setting, we present a variant of the well-studied EXP4 algorithm [6] (fully presented in Appendix A), that incorporates delay-robust loss estimators specialized to the CMAB setting. At a high-level, using direct access to $\Pi$, the algorithm performs multiplicative weight updates over the $N$-dimensional simplex $\Delta_N$, while using all of the feedback that arrives in each round $t$ to construct loss estimators, denoted by $\hat{c}_t \in \mathbb{R}_+^N$ and defined in Equation (2). These estimators are inspired by Jin et al. [20], and are reminiscent of the standard importance-weighted loss estimators, with an additional term in the denominator which induces an under-estimation bias and allows for a simplified analysis. Interestingly, these estimators exhibit a coupling between the context $x_t$, which arrives at a given round $t$, and the sampling distribution $p_{t+d_t}$ from a *future* round, and can be thought of as a mechanism that incentivizes actions whose sampling probability has increased between rounds $t$ and $t + d_t$, with respect to the context $x_t$. The main result for Algorithm 3 is given bellow.

**Theorem 3.1.** *Algorithm 3 attains expected regret bound of*

$$\mathbb{E}[\mathcal{R}_T] \le \frac{\log N}{\eta} + \eta KT + 2\eta D,$$

---

[0] In the function approximation setting we assume the delay sequence satisfies a FIFO property, see Section 4 for details.

*where the expectation is over the algorithm's randomness.*

*For $\eta = \sqrt{\frac{\log N}{KT+D}}$ we obtain an optimal bound of*

$$\mathbb{E}[\mathcal{R}_T] \le O\left(\sqrt{KT \log N} + \sqrt{D \log N}\right).$$

Intuitively, in the policy class setting we can directly optimize over $\Pi$ by using Hedge updates which exhibit stability properties that are crucial in the presence delayed feedback. Such stability properties are much harder to obtain in the function approximation setting where the contextual bandit algorithm does not have direct access to the policy class, and in particular is required to be computationally efficient; see Section 4 for details. We also remark that Algorithm 3 requires an upper bound on the sum of delays $D$, however, it can be made adaptive by utilizing a "doubling" mechanism as suggested in, e.g., [22]. The description of Algorithm 3 and the proof of Theorem 3.1 appear in Appendix A.

**Matching lower bound.** In Appendix C.2, we prove a matching lower bound showing that the upper bound obtained in Theorem 3.1 is optimal up to constant factors. To our knowledge, this is the first tight lower bound that applies to policy class learning with delayed feedback, as previous lower bounds (e.g., [10]) exhibit a delay dependence of $\sqrt{D}$ rather than $\sqrt{D \log N}$.

## 4  Online Function Approximation

In this section, we provide regret guarantees for CMAB with delayed feedback under the framework of online function approximation [13, 16]. In this setting, the learner has access to a class of loss functions $\mathcal{F} \subseteq \mathcal{X} \times \mathcal{A} \to [0, 1]$, where each function $f \in \mathcal{F}$ maps a context $x \in \mathcal{X}$ and an action $a \in \mathcal{A}$ to a loss $\ell \in [0, 1]$. We use $\mathcal{F}$ to approximate the context-dependent expected loss of any action $a \in \mathcal{A}$ for any context $x \in \mathcal{X}$. The CMAB algorithm can access $\mathcal{F}$ using an online least-squares regression (OLSR) oracle that will operate under the following standard realizability assumption.

**Assumption 4.1.** There exists a function $f_\star \in \mathcal{F}$ such that for all $(x, a) \in \mathcal{X} \times \mathcal{A}$, $f_\star(x, a) = \ell(x, a)$.

We assume access to a classical, non-delayed, online regression oracle with respect to the square loss function $h_{sq}(\hat{y}, y) = (\hat{y} - y)^2$. The oracle, which we denote by $\mathcal{O}_{\text{sq}}^{\mathcal{F}}$, is given as input at each round $t$ the past observations $(x_s, a_s, L_s(x_s, a_s))_{s=1}^{t-1}$ and outputs a function $\hat{f}_t : \mathcal{X} \times \mathcal{A} \to [0, 1]$. A general formulation of the online oracle model is discussed in Foster and Rakhlin [13]. We make use of the following standard online least-squares expected regret assumption of the oracle:

**Assumption 4.2** (Least-Squares Oracle Regret). The oracle $\mathcal{O}_{\text{sq}}^{\mathcal{F}}$ guarantees that for every sequence $\{(x_t, a_t, L_t)\}_{t=1}^T$ where $a_t \sim p_t$ and $L_t \in [0, 1]$ has $\mathbb{E}[L_t | x_t, a_t] = \ell(x_t, a_t)$, the expected least-squares regret is bounded as $\sum_{t=1}^T \mathbb{E}[(\hat{f}_t(x_t, a_t) - \ell(x_t, a_t))^2] \le \mathcal{R}_T(\mathcal{O}_{\text{sq}}^{\mathcal{F}})$.

Note that a stronger high-probability version of Assumption 4.2 is made in Foster and Rakhlin [13] in order to prove high probability regret bounds for CMAB, but for our use the weaker version suffices. Assumption 4.1 and Assumption 4.2 (or variants of it for other loss functions) are necessary to derive regret bounds for adversarial CMAB and are extensively used literature (see e.g., [13, 14]). However, a general implementation of online least-square regression oracle for a function class might be unstable. To justify the importance of stability, we show in the following result (proven in Appendix C.1) that Assumption 4.1 and Assumption 4.2 alone do not suffice for sublinear regret with function approximation in the presence of delays.

**Theorem 4.3.** *For any CMAB algorithm* ALG *in the function approximation setting there exists a contextual bandit instance with fixed delay $d = 1$ over a realizable loss class $\mathcal{F}$ with $|\mathcal{F}| = T + 1$ and an online oracle $\mathcal{O}_{sq}^{\mathcal{F}}$ satisfying $\mathcal{R}_T(\mathcal{O}_{sq}^{\mathcal{F}}) = 0$, on which* ALG *attains regret $\mathcal{R}_T = \Omega(T)$.*

Hence, we impose the following stability assumption on the oracle.

**Assumption 4.4** ($\beta$-stability). Let $\hat{f}_1, \hat{f}_2, \ldots, \hat{f}_T$ denote the function sequence outputted by the non-delayed oracle $\mathcal{O}_{\text{sq}}^{\mathcal{F}}$ on the observation sequence $\{(x_t, a_t, L_t)\}_{t=1}^T$. We assume that for some $\beta > 0$, it holds that $\mathbb{E}[\sum_{t=1}^T \|\hat{f}_t - \hat{f}_{t+1}\|_\infty^2] \le \beta$, where in $\|\cdot\|_\infty$ we take supremum over $x \in \mathcal{X}$ and $a \in \mathcal{A}$ and the expectation is over the loss realizations $\{L_t\}_{t=1}^T$.

We denote a $\beta$-stable OLSR oracle for the function class $\mathcal{F}$ by $\mathcal{O}_{\text{sq}}^{\mathcal{F},\beta}$. As a specific example of such an oracle, we present a Hedge-based version of Vovk's aggregating forecaster (Algorithm 2) and prove that it guarantees least-squares regret of $O(\log|\mathcal{F}|)$ while simultaneously satisfying Assumption 4.4 with $\beta = O(\log|\mathcal{F}|)$, which we use to derive an expected regret bound for this setting.

Our use of a non-delayed OLSR oracle to handle delayed CMAB setup with function approximation requires us to make an assumption on the delay sequence, namely, that observations arrive in FIFO order.[1] This is formalized in the following assumption and further explained in the following.

**Assumption 4.5** (FIFO). We assume the delay sequence $(d_1, \ldots, d_T)$ satisfies $s + d_s \leq t + d_t$ whenever $1 \leq s \leq t \leq T$. In particular, if the observation from time $t$ does not arrive (that is, $t + d_t > T$) then neither do all observations from rounds $t' > t$.

## 4.1 Algorithm: Delay-Adapted Function Approximation for CMAB

We present Algorithm DA-FA (Algorithm 1) for regret minimization in CMAB with delayed feedback for the function approximation framework. Algorithm 1 essentially uses the most up-to-date approximation of the loss until delayed observations arrive. When they arrive, the algorithm feeds them to the oracle one by one, ignores the midway approximations, and uses only the newest loss approximation. In each round $t = 1, 2, \ldots, T$ the algorithm operates as follows. Let $\alpha(t) < t$ denote the number of observations that arrived at round $t$. Denote these observations by $\{(s_i^t, L(x_{s_i^t}, a_{s_i^t}))\}_{i=1}^{\alpha(t)}$, where $s_1^t \leq \ldots \leq s_{\alpha(t)}^t$ denote the time steps of the non-delayed related context and action associated with these delayed loss observations. It then holds that $s_i^t + d_{s_i^t} = t$ for all $i \in [\alpha(t)]$. Note that we assume that the delayed observations arrive in FIFO order, meaning that the delayed observation from round $\tau$ always arrives before (or in parallel to) that of round $\tau + 1$ for all $\tau \in [T]$. Then, for $i = 1, \ldots, \alpha(t)$, we feed the oracle with the example $(x_{s_i^t}, a_{s_i^t}, L(x_{s_i^t}, a_{s_i^t}))$ by order and observe the predicted function $\hat{f}_{t - d_{s_i^t}}$. Let $\tau^t = t - d_{s_{\alpha(t)}^t} = s_{\alpha(t)}^t$ denote the index of the last observed delayed loss. After processing all the data that arrived, the current context $x_t$ is revealed and the algorithm uses the last predicted function $\hat{f}_{\tau^t}$ to solve the regularized convex optimization problem specified in Equation (1), and plays an action sampled from the resulted distribution.

---

**Algorithm 1** Delay-Adapted Function Approximation for CMAB (DA-FA)

1: **inputs:** Function class $\mathcal{F}$ for loss approximation, learning rate $\gamma$, $\beta$-stable OLSR oracle $\mathcal{O}_{\text{sq}}^{\mathcal{F},\beta}$.
2: **for** round $t = 1, \ldots, T$ **do**
3:     observe $\alpha(t) < t$ losses $\{(s_i^t, L(x_{s_i^t}, a_{s_i^t}))\}_{i=1}^{\alpha(t)}$ where $\forall i \in [\alpha(t)]$, $s_i^t + d_{s_i^t} = t$ and $s_1^t \leq \ldots \leq s_{\alpha(t)}^t$.
4:     **for** $i = 1, 2, \ldots, \alpha(t)$ **do**
5:         update $\mathcal{O}_{\text{sq}}^{\mathcal{F},\beta}$ with the example $((x_{s_i^t}, a_{s_i^t}), L(x_{s_i^t}, a_{s_i^t}))$.
6:         observe the oracle's output $\hat{f}_{t - d_{s_i^t}} \leftarrow \mathcal{O}_{\text{sq}}^{\mathcal{F},\beta}$.
7:     let $\tau^t := t - d_{s_{\alpha(t)}^t} = s_{\alpha(t)}^t$ denote index of the last observed loss.
8:     use $\hat{f}_{\tau^t}$ as the current loss approximation.
9:     observe context $x_t \in \mathcal{X}$.
10:    solve

$$p_t \in \underset{p \in \Delta_{\mathcal{A}}}{\arg\min} \sum_{a \in \mathcal{A}} p(a) \hat{f}_{\tau^t}(x_t, a) - \frac{1}{\gamma} \sum_{a \in \mathcal{A}} \log(p(a)). \tag{1}$$

11:    play the action $a_t$ sampled from $p_t$

---

The regret bound for Algorithm 1 is given in the following.

**Theorem 4.6.** *Let* $\gamma = \sqrt{KT / \mathcal{R}_T \mathcal{O}_{\text{sq}}^{\mathcal{F},\beta}}$. *Then Algorithm 1 has an expected regret bound of*

$$\mathbb{E}[\mathcal{R}_T] \leq O\left(\sqrt{KT \mathcal{R}_T(\mathcal{O}_{\text{sq}}^{\mathcal{F},\beta})} + \sqrt{d_{\max} D \beta}\right).$$

---

[1] In particular, this includes the case of fixed delay $d$.

*In particular, this implies the expected regret is bounded as*

$$\mathbb{E}[\mathcal{R}_T] \leq O\left( \sqrt{KT\mathcal{R}_T(\mathcal{O}_{sq}^{\mathcal{F},\beta})} + D^{3/4}\sqrt{\beta} \right).$$

We remark that Theorem 4.6 actually holds with high probability whenever Assumption 4.2 and Assumption 4.4 hold in high probability rather than in expectation.

**Why FIFO order is needed.** In the following analysis, we make use of the assumption that the observations arrive in FIFO order to argue that the realized functions that the oracle outputs throughout the process correspond to functions that are outputs of the oracle on the non-delayed observation sequence. Otherwise, the delay can cause a permutation in the order of observations, inducing a sequence of realized outputs that might be different than those of the non-delayed oracle, in which case we cannot relate the realized regret to the regret of the oracle on the non-delayed sequence.

**Computational efficiency.** The optimization problem in Equation (1) is convex and can be solved efficiently to arbitrary precision. Thus Algorithm 1 is clearly efficient, assuming an efficient oracle.

## 4.2 Analysis

In this subsection, we analyze Algorithm 1, proving Theorem 4.6. Our main technical challenge is reflected in the regret decomposition. As in all previous literature regarding delayed feedback, the main challenge is to derive a bound where the sum of delays $D$ is separated from the number of actions $K$. Usually, this separation is obtained by an appropriate choice of loss estimators. In our case, however, the loss is estimated by the oracle, and hence not transparent to the algorithm. Our way to create the desired separation is via the regret decomposition described in the following. Let $\{\hat{f}_1, \hat{f}_2, \ldots, \hat{f}_T\}$ denote the functions predicted by the OLSR oracle on the non-delayed observation sequence $\{(x_1, a_1, L(x_1, a_1)), \ldots, (x_T, a_T, L(x_T, a_T))\}$. That is, $\hat{f}_{i+1} = \mathcal{O}_{sq}^{\mathcal{F},\eta}(\cdot; (x_1, a_1, L(x_1, a_1)), \ldots, (x_i, a_i, L(x_i, a_i))), \ \forall i \in [T-1]$. For convenience, we denote the optimal (randomized) policy by $p_\star(\cdot|x)$ for all $x \in \mathcal{X}$. Then, the regret is given by $\mathcal{R}_T = \sum_{t=1}^{T}(p_t - p_\star(\cdot \mid x_t)) \cdot \ell(x_t, \cdot)$ and can be decomposed and bounded as follows.

$$\mathcal{R}_T \leq \underbrace{d_{\max}}_{(a)} + \underbrace{\sum_{t=d_{\max}+1}^{T}(p_t - p_\star(\cdot \mid x_t)) \cdot \hat{f}_{\tau^t}(x_t, \cdot)}_{(b)} + \underbrace{\sum_{t=d_{\max}+1}^{T} p_t \cdot \left( \ell(x_t, \cdot) - \hat{f}_t(x_t, \cdot) \right)}_{(c)}$$

$$+ \underbrace{\sum_{t=d_{\max}+1}^{T} p_\star(\cdot \mid x_t) \cdot \left( \hat{f}_t(x_t, \cdot) - \ell(x_t, \cdot) \right)}_{(d)} + \underbrace{\sum_{t=d_{\max}+1}^{T}(p_t - p_\star(\cdot \mid x_t)) \cdot \left( \hat{f}_t(x_t, \cdot) - \hat{f}_{\tau^t}(x_t, \cdot) \right)}_{(e)}$$

In the above decomposition, term $(a)$ is the regret on the first $d_{\max}$ steps and hence bounded trivially by $d_{\max}$. Term $(b)$ is the regret with respect to the approximated delayed loss. Term $(c)$ is the approximation error with respect to the policy induced by $p_t$ when considering the non-delayed approximated loss, and will be bounded by the oracle's regret. Term $(d)$ is the approximation error with respect to the optimal $p_\star(\cdot|\cdot)$ when considering the non-delayed approximated loss. We remark that the Assumption 4.5 is used when bounding terms $(c)$ and $(d)$. Lastly, term $(e)$ is the regret caused by the delay drift in approximation. This term will be shown to be bounded by $\sqrt{d_{\max}D\beta}$, with no direct dependence on the number of actions $K$. We bound each term individually in the following lemmas, and then combine the results to conclude Theorem 4.6. We begin with term $(b)$, whose bound follows from first-order optimality conditions for convex optimization.

**Lemma 4.7** (Term (b) bound)**.** *It holds true that*

$$\sum_{t=d_{\max}+1}^{T}(p_t(\cdot) - p_\star(\cdot|x_t)) \cdot \hat{f}_{\tau^t}(x_t, \cdot) \leq \frac{KT}{\gamma} - \sum_{t=d_{\max}+1}^{T} \sum_{a \in \mathcal{A}} \frac{p_\star(a|x_t)}{\gamma p_t(a)}.$$

Term $(c)$ is bounded using the AM-GM inequality, and applying Assumption 4.2.

**Lemma 4.8** (Term (c) bound). *It holds that*

$$\mathbb{E}\left[\sum_{t=d_{\max}+1}^{T} p_t \cdot (\ell(x_t, \cdot) - \hat{f}_t(x_t, \cdot))\right] \leq \frac{KT}{\gamma} + \gamma \mathcal{R}_T(\mathcal{O}_{sq}^{\mathcal{F},\beta}).$$

Term $(d)$ is bounded using the AM-GM inequality to change the measure from $p_\star(\cdot|x_t)$ to $p_t$, to then apply the non-delayed oracle's regret bound.

**Lemma 4.9** (Term (d) bound). *The following holds true*

$$\mathbb{E}\left[\sum_{t=d_{\max}+1}^{T} p_\star(\cdot|x_t) \cdot (\hat{f}_t(x_t, \cdot) - \ell(x_t, \cdot))\right] \leq \sum_{t=d_{\max}+1}^{T} \sum_{a \in \mathcal{A}} \frac{p_\star(a|x_t)}{\gamma p_t(a)} + \gamma \mathcal{R}_T(\mathcal{O}_{sq}^{\mathcal{F},\beta}).$$

The proofs of Lemmas Lemmas 4.7 to 4.9 are inspired by those of Levy et al. [26], and included for completeness in Appendix B.1.

Lastly, we bound the delay-dependent term $(e)$. This is where we need to make use of the oracle's stability given in Assumption 4.4 in order to obtain the bound given in the following lemma, whose full proof can also be found in Appendix B.1.

**Lemma 4.10** (Term (e) bound). *Under Assumption 4.4, the following holds true.*

$$\mathbb{E}\left[\sum_{t=d_{\max}+1}^{T} (p_t - p_\star(\cdot \mid x_t)) \cdot (\hat{f}_t(x_t, \cdot) - \hat{f}_{\tau^t}(x_t, \cdot))\right] \leq 2\sqrt{d_{\max}D\beta}.$$

*Proof sketch.* Using Hölder's inequality, it holds that

$$\sum_{t=d_{\max}+1}^{T} (p_t - p_\star(\cdot \mid x_t)) \cdot (\hat{f}_t(x_t, \cdot) - \hat{f}_{\tau^t}(x_t, \cdot)) \leq 2 \sum_{t=d_{\max}+1}^{T} \sum_{i=1}^{d_{\tau^t}} \|\hat{f}_{t-i} - \hat{f}_{t-(i-1)}\|_\infty$$

$$\leq 2 \sum_{t=d_{\max}+1}^{T} \sigma_t \|\hat{f}_t - \hat{f}_{t+1}\|_\infty,$$

where $\sigma_t$ is the number of pending observations (that is, which have not yet arrived) as of round $t$. Taking expectation while using Jensen's inequality, the Cauchy-Schwarz inequality and Assumption 4.4, the above can be further bounded by

$$\leq 2\sqrt{\left(\sum_{t=d_{\max}+1}^{T} \sigma_t^2\right) \cdot \left(\sum_{t=d_{\max}+1}^{T} \mathbb{E}\left[\|\hat{f}_t - \hat{f}_{t+1}\|_\infty^2\right]\right)} \leq 2\sqrt{d_{\max}D\beta},$$

where we used that $\sigma_t \leq d_{max}$ and $\sum_t \sigma_t = D$. $\qquad\square$

We now have what we need to prove Theorem 4.6.

*Proof of Theorem 4.6.* Putting Lemmas 4.7 to 4.10 together, the expected regret of Algorithm 1 bounded by $\mathbb{E}[\mathcal{R}_T] \leq d_{\max} + 2\frac{KT}{\gamma} + 2\gamma \mathcal{R}_T(\mathcal{O}_{sq}^{\mathcal{F},\beta}) + 2\sqrt{d_{\max}D\beta}$. Choosing $\gamma = \sqrt{\frac{KT}{\mathcal{R}_T(\mathcal{O}_{sq}^{\mathcal{F},\beta})}}$ yields the bound.

To prove the second statement of the theorem, we note that whenever an observation from given round $t$ arrives with the maximal delay $d_{\max}$, Assumption 4.5 implies that all observations from rounds $t' \in \{t+1, \ldots, t + d_{\max} - 1\}$ arrive after round $t + d_{\max}$. Therefore, we can lower bound the sum of delays as a function of $d_{\max}$ as

$$D \geq \sum_{t'=t}^{t+d_{\max}} d_{t'} \geq \sum_{t'=t}^{t+d_{\max}} (d_{\max} - (t' - t)) = \Omega(d_{\max}^2).$$

Therefore, $d_{\max} = O(\sqrt{D})$ and the bound follows. $\qquad\square$

### 4.3 Stability analysis of Hedge-based Vovk's aggregating forecaster

In this section, we present a concrete online least squares regression oracle implementation that enjoys an expected square-loss regret bound of $O(\log|\mathcal{F}|)$ while simultaneously satisfying Assumption 4.4 with $\beta \lesssim \log|\mathcal{F}|$ (see Corollary 4.13 and Lemma 4.12). We then use this result to derive an expected regret bound of $O(\sqrt{KT\log|\mathcal{F}|} + \sqrt{d_{\max}D\log|\mathcal{F}|})$ for Algorithm 1 using this oracle. The oracle is a Hedge-based version of Vovk's aggregating forecaster [39] applied for the square loss (see Algorithm 2 for details). Interestingly, even though Algorithm 2 uses a constant (that is, large) step size, its square-loss regret is independent of $T$, and perhaps more surprisingly, the expected sum of KL-deviations between consecutive iterates $q_t$ and $q_{t+1}$ is also independent of $T$. This crucial property allows us to use this general purpose oracle in order to obtain a non-trivial regret bound in the general function approximation setting.

**Hedge-based version of Vovk's aggregating forecaster.** Algorithm 2 performes Hedge updates over

---

**Algorithm 2** Hedge-based Vovk's aggregating forecaster

---
1: **parameters:** (finite) function class $\mathcal{F} \subseteq \mathcal{X} \times \mathcal{A} \to [0,1]$, step size $\eta > 0$
2: Initialize $q_1 \in \Delta_{\mathcal{F}}$ as the uniform distribution over $\mathcal{F}$.
3: **for** round $t = 1, 2, \ldots, T$: **do**
4:     Return $\hat{f}_t = \sum_{f \in \mathcal{F}} q_t(f)f$ to contextual bandit algorithm.
5:     Observe feedback $z_t = (x_t, a_t)$ and realized loss $y_t \in [0,1]$.
6:     Update $q_t$ as follows: $q_{t+1}(f) \propto q_t(f)e^{-\eta(f(z_t)-y_t)^2}, \quad \forall f \in \mathcal{F}$.

---

the finite function class $\mathcal{F}$ using the squared loss. That is, it maintains a distribution over functions $q_t \in \Delta(\mathcal{F})$ and in each round $t$ returns an aggregation of the functions in $\mathcal{F}$ by the weights $q_t$.

In the next theorem, we prove an expected regret bound and use it to establish stability guarantees for Algorithm 2. We emphasize that while regret analyses of Vovk's aggregating forecaster exist in the literature (e.g. [9]), the resulting stability property is novel, and in particular only holds under realizability. The full proof appears in Appendix B.2.

**Theorem 4.11** (Regret and stability guarantee for finite function classes). *For $t \in [T]$ denote by $q_t \in \Delta(\mathcal{F})$ the probability measure over functions in $\mathcal{F}$ computed by Algorithm 2 at time step $t$.*

*Then, the following holds for any $\eta \leq 1/18$:*
*(1) Expected regret:*

$$\sum_t \mathbb{E}[(\hat{f}_t(z_t) - f_\star(z_t))^2] \leq \frac{2\log|\mathcal{F}|}{\eta}.$$

*(2) Stability:*

$$\sum_t \mathbb{E}[KL(q_t\|q_{t+1})] \leq \log|\mathcal{F}|,$$

*where the expectation is taken over the loss realizations $y_1, \ldots, y_T$.*

**Regret bound.** We use the latter result to derive an oracle-specific regret bound for finite and realizable function classes. Our regret bound, stated in Corollary 4.13, follows from Theorem 4.11 using Lemma 4.12, which proves that Algorithm 2 with constant step size satisfies Assumption 4.4 for $\beta = O(\log|\mathcal{F}|)$ (see the next lemma). Applying this result to Theorem 4.6 yields the bound.

**Lemma 4.12** (Stability of Algorithm 2). *Under Assumption 4.1, for a finite function class $\mathcal{F}$, Algorithm 2 with step size $\eta = 1/18$ satisfies Assumption 4.4 with $\beta = 2\log|\mathcal{F}|$.*

*Proof.* Using the form of $\hat{f}_1, \ldots, \hat{f}_T$, the outputs of Algorithm 2, for all $t \in [T]$ it holds that

$$\|\hat{f}_t - \hat{f}_{t+1}\|_\infty = \|\sum_{f \in \mathcal{F}}(q_t(f) - q_{t+1}(f))f\|_\infty \leq \sum_{f \in \mathcal{F}}|q_t(f) - q_{t+1}(f)|\|f\|_\infty$$

$$\leq \|q_t - q_{t+1}\|_1 \leq \sqrt{2} \cdot \sqrt{KL(q_t\|q_{t+1})},$$

where we used the fact that $\|f\|_\infty \leq 1$ for all $f \in \mathcal{F}$ and Pinsker's inequality. Taking a square, summing over $t$ and taking expectations we get,

$$\sum_{t=1}^T \mathbb{E}[\|\hat{f}_t - \hat{f}_{t+1}\|_\infty^2] \leq 2 \sum_{t=1}^T \mathbb{E}[KL(q_t\|q_{t+1})] \leq 2\log|\mathcal{F}|$$

where we have used the second result of Corollary 4.13. □

We can now immediately obtain the following expected regret bound for Algorithm 1 when used with Algorithm 2 as a square-loss oracle.

**Corollary 4.13.** *Let $\mathcal{F}$ denote a realizable loss function class, where $|\mathcal{F}| < \infty$. Suppose we use Algorithm 2 as an oracle implementation for online least-square regression with $\eta = 1/18$ and choose $\gamma = \sqrt{\frac{KT}{36\log|\mathcal{F}|}}$ for Algorithm 1. Then,*

$$\mathbb{E}[\mathcal{R}_T] \leq O\Big(\sqrt{KT\log|\mathcal{F}|} + \sqrt{d_{\max}D\log|\mathcal{F}|}\Big).$$

**Lower bound.** In Corollary C.5 of Appendix C.2 we state and prove the first lower bound of

$$\Omega(\sqrt{KT\log|\mathcal{F}|} + \sqrt{D\log|\mathcal{F}|})$$

for CMAB under delayed feedback assuming realizable function approximation. The lower bound implies that our result in Corollary 4.13 is far by a $\sqrt{d_{\max}}$ factor from the optimal regret bound.

## 5   Conclusions and Discussion

In this paper we presented regret minimization algorithms for adversarial CMAB with delayed feedback, where both the contexts and delays are chosen by a possibly adaptive adversary. We considered the problem under the two mainstream frameworks for adversarial CMAB learning: online function approximation and policy class learning.

For the policy class learning setup, we presented Algorithm 3 and proved that it obtains an expected regret bound that is optimal up to logarithmic factors.

For online function approximation, we presented Algorithm 1 and analyzed its regret under a stability assumption related to the online regression oracle in use, which affects the delay-dependent term of our bound. Additionally, we analyzed the expected regret of a version of Vovk's aggregating forecaster and shown it satisfies the required stability guarantees, allowing us to obtain a nontrivial expected regret bound of $O(\sqrt{KT\log(|\mathcal{F}|)} + \sqrt{d_{\max}D\log|\mathcal{F}|})$, which is optimal up to $\sqrt{d_{\max}}$.

Our work leaves some open questions that we believe are very interesting for future research. One possible direction is to remove the $\sqrt{d_{\max}}$ factor from the delay-dependent term in our bound, which is presumably sub-optimal. Furthermore, as our lower bound in Theorem 4.3 shows, without any assumption on the oracle in use other than Assumption 4.2, linear regret is unavoidable. We therefore find it interesting to investigate different or weaker assumptions on the oracle that enable non-trivial regret guarantees.

## Acknowledgements

We would like to thank the reviewers for their helpful comments.

This project has received funding from the European Research Council (ERC) under the European Union's Horizon 2020 research and innovation program (grant agreement No. 882396 and grant agreement No. 101078075). Views and opinions expressed are however those of the author(s) only and do not necessarily reflect those of the European Union or the European Research Council. Neither the European Union nor the granting authority can be held responsible for them. This work received additional support from the Israel Science Foundation (ISF, grant numbers 993/17 and 2549/19), Tel Aviv University Center for AI and Data Science (TAD), the Yandex Initiative for Machine Learning at Tel Aviv University, the Len Blavatnik and the Blavatnik Family Foundation.

AC is supported by the Israeli Science Foundation (ISF) grant no. 2250/22.

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

# A    Proofs for Section 3

In this section, we analyze the regret of Algorithm 3 in the policy class setting and prove Theorem 3.1.

---

**Algorithm 3** EXP4 with Delay-Adapted Loss Estimators (EXP4-DALE)

---

1: **inputs:**
   - Finite policy class $\Pi \subseteq \mathcal{X} \to \mathcal{A}$ with $|\Pi| = N$,
   - Upper bound on the sum of delays, $D$.
   - Step size $\eta > 0$.
2: Initialize $p_1 \in \Delta_N$ as the uniform distribution over $\Pi$.
3: **for** round $t = 1, \ldots, T$ **do**
4:     Receive context $x_t \in \mathcal{X}$.
5:     Sample $\pi \sim p_t$ and play $a_t = \pi(x_t)$.
6:     Observe feedback $(s, L(x_s, a_s))$ for all $s \leq t$ with $s + d_s = t$ and construct loss estimators

$$\hat{c}_{s,i} = \frac{L(x_s, a_s)\mathbb{I}[\pi_i(x_s) = a_s]}{\max\left\{Q_{s,a_s}, \tilde{Q}^t_{s,a_s}\right\}} \quad \forall i \in [N], \tag{2}$$

   where we define $Q_{s,a} = \sum_{i=1}^N p_{s,i}\mathbb{I}[\pi_i(x_s) = a]$ and $\tilde{Q}^t_{s,a} = \sum_{i=1}^N p_{t,i}\mathbb{I}[\pi_i(x_s) = a]$.
7:     Update

$$p_{t+1,i} \propto p_{t,i} \exp\left(-\eta \sum_{s:s+d_s=t} \hat{c}_{s,i}\right). \tag{3}$$

---

Throughout this section, we use the notation $\mathbb{E}_t[\cdot]$ to denote an expectation conditioned on the entire history up to round $t$. We define the standard (unbiased) importance-weighted loss estimators by

$$\tilde{c}_{t,i} = \frac{L(x_t, a_t)\mathbb{I}[\pi_i(x_t) = a_t]}{Q_{t,a_t}} \quad \forall i \in [N], \tag{4}$$

**Theorem A.1.** *Algorithm 3 attains the following expected regret bound:*

$$\mathbb{E}[\mathcal{R}_T] \leq \frac{\log N}{\eta} + \frac{\eta}{2}\mathbb{E}\left[\sum_{t=1}^T \sum_{i=1}^N p_{t+d_t,i}\hat{c}_{t,i}^2\right] + 2\mathbb{E}\left[\sum_{t=1}^T \|p_{t+d_t} - p_t\|_1\right].$$

*Proof.* The regret may be decomposed as follows:

$$\mathcal{R}_T = \sum_{t=1}^T c_t \cdot (p_t - p^\star)$$

$$= \underbrace{\sum_{t=1}^T p_t \cdot (c_t - \hat{c}_t)}_{Bias_1} + \underbrace{\sum_{t=1}^T p^\star \cdot (\hat{c}_t - c_t)}_{Bias_2} + \underbrace{\sum_{t=1}^T (p_t - p_{t+d_t}) \cdot \hat{c}_t}_{Drift} + \underbrace{\sum_{t=1}^T (p_{t+d_t} - p^\star) \cdot \hat{c}_t}_{OMD}, \tag{5}$$

where $c_{t,i} = L(x_t, \pi_i(x_t))$ for $i \in [N]$. The $OMD$ term can be bounded by referring to Lemma 9 of [35] which asserts that

$$\sum_{t=1}^T (p_{t+d_t} - p^\star) \cdot \hat{c}_t \leq \frac{\log N}{\eta} + \frac{\eta}{2}\sum_{t=1}^T \sum_{i=1}^{|\Pi|} p_{t+d_t,i}\hat{c}_{t,i}^2. \tag{6}$$

while noting that this lemma does not require a specific form of loss estimators, only that they are nonnegative, as is the case for our delay-adapted estimators defined in Equation (2). We also note that the $Bias_2$ term is non-positive in expectation, since the delay-adapted estimators satisfy $\mathbb{E}_t[\hat{c}_{t,i}] \leq c_{t,i}$ for $i \in [N]$. Thus, to conclude the proof we are left with bounding the $Drift$ and $Bias_1$ terms, whose bounds are given in Lemma A.2 and Lemma A.3 that follow. $\square$

*Proof of Theorem 3.1.* First, we show that

$$\mathbb{E}\left[\sum_t \sum_i p_{t+d_t,i}\hat{c}_{t,i}^2\right] \le KT.$$

Indeed, using the definition of the delay-adapted loss estimators $\hat{c}_t$, it holds that

$$
\begin{aligned}
\mathbb{E}\left[\sum_t \sum_i p_{t+d_t,i}\hat{c}_{t,i}^2\right] &= \mathbb{E}\left[\sum_t \sum_i p_{t+d_t,i}\left(\frac{L(x_t,a_t)\mathbb{I}[\pi_i(x_t)=a_t]}{\max\left\{Q_{t,a_t},\tilde{Q}_{t,a_t}^{t+d_t}\right\}}\right)^2\right] \\
&\le \mathbb{E}\left[\sum_t \frac{1}{\tilde{Q}_{t,a_t}^{t+d_t}}\sum_i \frac{p_{t+d_t,i}\mathbb{I}[\pi_i(x_t)=a_t]}{Q_{t,a_t}}\right] \\
&= \mathbb{E}\left[\sum_t \frac{1}{Q_{t,a_t}}\right] = \mathbb{E}\left[\sum_t \sum_a \frac{Q_{t,a}}{Q_{t,a}}\right] = KT.
\end{aligned}
$$

Thus, using Theorem A.1 together with Lemma A.4 gives the bound claimed in Theorem 3.1. $\qquad\square$

**Lemma A.2** (Bounding the Drift term). *The $Drift$ term given in Equation (5) is bounded in expectation as follows:*

$$\mathbb{E}\left[\sum_{t=1}^T (p_t - p_{t+d_t})\cdot\hat{c}_t\right] \le \mathbb{E}\left[\sum_{t=1}^T \|p_t - p_{t+d_t}\|_1\right].$$

*Proof.* First, we note that the delay-adapted loss estimators $\hat{c}_t$ are upper-bounded by the standard, conditionally unbiased importance-weighted estimators $\tilde{c}_t$ defined in Equation (4). Therefore, we can bound the $Drift$ term as follows:

$$
\begin{aligned}
\mathbb{E}\left[\sum_{t=1}^T (p_t - p_{t+d_t})\cdot\hat{c}_t\right] &\le \mathbb{E}\left[\sum_{t=1}^T \sum_{i=1}^N |p_{t,i} - p_{t+d_t,i}|\hat{c}_{t,i}\right] \\
&\le \mathbb{E}\left[\sum_{t=1}^T \sum_{i=1}^N |p_{t,i} - p_{t+d_t,i}|\tilde{c}_{t,i}\right] \\
&= \mathbb{E}\left[\sum_{t=1}^T \sum_{i=1}^N |p_{t,i} - p_{t+d_t,i}|\cdot c_{t,i}\right] \\
&\le \mathbb{E}\left[\sum_{t=1}^T \|p_t - p_{t+d_t}\|_1\right],
\end{aligned}
$$

where the last step follows from Hölder's inequality and the fact that $\|c_t\|_\infty \le 1$. $\qquad\square$

**Lemma A.3.** *The $Bias_1$ term given in Equation (5) is bounded in expectation as follows:*

$$\mathbb{E}\left[\sum_t p_t\cdot(c_t - \hat{c}_t)\right] \le \mathbb{E}\left[\sum_{t=1}^T \|p_t - p_{t+d_t}\|_1\right].$$

*Proof.* We note losses and loss estimators can be indexed by actions rather than policies and use the the notation $c_{t,a} = L(x_t,a)$ and $\hat{c}_{t,a} = \frac{c_{t,a}\mathbb{I}[a_t=a]}{M_{t,a}}$ where $M_{t,a} = \max\left\{Q_{t,a},\tilde{Q}_{t,a}^{t+d_t}\right\}$. Therefore, using the fact that $\mathbb{E}_t[\hat{c}_{t,a}] = c_{t,a}\frac{Q_{t,a}}{M_{t,a}}$, the $Bias_1$ term can be bounded as follows:

$$\mathbb{E}\left[\sum_{t=1}^{T} p_t \cdot (c_t - \hat{c}_t)\right] = \mathbb{E}\left[\sum_{t=1}^{T}\sum_{a=1}^{K} Q_{t,a}(L(x_t, a) - \hat{c}_{t,a})\right]$$

$$= \mathbb{E}\left[\sum_{t=1}^{T}\sum_{a=1}^{K} Q_{t,a}L(x_t, a)\left(1 - \frac{Q_{t,a}}{M_{t,a}}\right)\right]$$

$$\leq \mathbb{E}\left[\sum_{t=1}^{T}\sum_{a=1}^{K} \frac{Q_{t,a}}{M_{t,a}}(M_{t,a} - Q_{t,a})\right]$$

$$\leq \mathbb{E}\left[\sum_{t=1}^{T}\sum_{k=1}^{K}\left(\max\left\{Q_{t,a}, \tilde{Q}_{t,a}^{t+d_t}\right\} - Q_{t,a}\right)\right]$$

$$\leq \mathbb{E}\left[\sum_{t=1}^{T}\sum_{a=1}^{K}\left|\tilde{Q}_{t,a}^{t+d_t} - Q_{t,a}\right|\right].$$

Now, by the definition of $Q_{t,a}$, $\tilde{Q}_{t,a}^{t+d_t}$ and the triangle inequality, we have

$$\mathbb{E}\left[\sum_{t=1}^{T}\sum_{a=1}^{K}\left|\tilde{Q}_{t,a}^{t+d_t} - Q_{t,a}\right|\right] \leq \mathbb{E}\left[\sum_{t=1}^{T}\sum_{a=1}^{K}\sum_{i:\pi_i(x_t)=a}|p_{t+d_t,i} - p_{t,i}|\right] = \mathbb{E}\left[\sum_{t=1}^{T}\|p_t - p_{t+d_t}\|_1\right],$$

concluding the proof. $\qquad\square$

**Lemma A.4** (Distribution drift). *The following holds for the iterates $\{p_t\}_{t=1}^{T}$ of Algorithm 3:*

$$\mathbb{E}\left[\sum_{t=1}^{T}\|p_{t+d_t} - p_t\|_1\right] \leq \eta(D + T).$$

*Proof.* Define

$$F_t(p) = p \cdot \sum_{s:s+d_s<t}\hat{c}_s + \frac{1}{\eta}R(p),$$

where $R(p) = \sum_{i=1}^{N} p_i \log p_i$, so that $p_t = \arg\min_{p\in\Delta_\Pi} F_t(p)$. Note that $R(\cdot)$ is 1-strongly convex with respect to $\|\cdot\|_1$, and therefore $F_t(\cdot)$ are $1/\eta$-strongly convex. Thus, using first-order optimality conditions for $p_t$ and $p_{t+1}$, we have:

$$F_t(p_{t+1}) \geq F_t(p_t) + \nabla F_t(p_t) \cdot (p_{t+1} - p_t) + \frac{1}{2\eta}\|p_{t+1} - p_t\|_1^2 \geq F_t(p_t) + \frac{1}{2\eta}\|p_{t+1} - p_t\|_1^2,$$

$$F_{t+1}(p_t) \geq F_{t+1}(p_{t+1}) + \nabla F_{t+1}(p_{t+1}) \cdot (p_t - p_{t+1}) + \frac{1}{2\eta}\|p_{t+1} - p_t\|_1^2 \geq F_{t+1}(p_{t+1}) + \frac{1}{2\eta}\|p_{t+1} - p_t\|_1^2.$$

Summing the two inequalities, we obtain

$$\frac{1}{\eta}\|p_{t+1} - p_t\|_1^2 \leq F_{t+1}(p_t) - F_t(p_t) + F_t(p_{t+1}) - F_{t+1}(p_{t+1})$$

$$= \left(\sum_{s:s+d_s=t}\hat{c}_s\right) \cdot (p_t - p_{t+1})$$

$$\leq \sum_i\left(\sum_{s:s+d_s=t}\hat{c}_{s,i}\right)|p_{t,i} - p_{t+1,i}|$$

$$\leq \sum_i\left(\sum_{s:s+d_s=t}\tilde{c}_{s,i}\right)|p_{t,i} - p_{t+1,i}|,$$

where $\tilde{c}_{s,i}$ are the standard (unbiased) importance-weighted loss estimators. Taking expectations while using $\mathbb{E}[(\cdot)^2] \geq (\mathbb{E}[\cdot])^2$ and Hölder's inequality, we obtain

$$\frac{1}{\eta}(\mathbb{E}\|p_{t+1} - p_t\|_1)^2 \leq \frac{1}{\eta}\mathbb{E}\Big[\|p_{t+1} - p_t\|_1^2\Big]$$

$$\leq \mathbb{E}\Bigg[\sum_i \Bigg(\sum_{s:s+d_s=t} c_{s,i}\Bigg) \cdot |p_{t+1,i} - p_{t,i}|\Bigg]$$

$$\leq m_t \mathbb{E}\|p_{t+1} - p_t\|_1,$$

where $m_t = |\{s : s + d_s = t\}|$ is the number of observations that arrive on round $t$. Dividing through by the right-hand side of the inequality above, we obtain

$$\mathbb{E}\|p_{t+1} - p_t\|_1 \leq \eta m_t,$$

and using the triangle inequality we have

$$\mathbb{E}\|p_{t+d_t} - p_t\|_1 \leq \sum_{s=1}^{d_t} \mathbb{E}\|p_{t+s} - p_{t+s-1}\|_1 \leq \eta \sum_{s=1}^{d_t} m_{t+s-1} = \eta M_{t,d_t},$$

where $M_{t,d_t}$ is the number of observations that arrive between rounds $t$ and $t + d_t - 1$. Using Lemma C.7 in [20], we conclude the proof via

$$\mathbb{E}\Bigg[\sum_{t=1}^{T}\|p_{t+d_t} - p_t\|_1\Bigg] \leq \eta \sum_{t=1}^{T} M_{t,d_t} \leq \eta(D + T).$$

$\square$

# B Proofs for Section 4

## B.1 Proofs for Subsection 4.2

In this subsection, we provide the proofs of the lemmas required to derive regret guarantees for algorithm DA-FA (Algorithm 1), proving Theorem 4.6.

Consider the following regret decomposition,

$$
\mathcal{R}_T = \sum_{t=1}^{d_{\max}} (p_t - p_\star(\cdot \mid x_t)) \cdot \ell(x_t, \cdot) + \sum_{t=d_{\max}+1}^{T} (p_t - p_\star(\cdot \mid x_t)) \cdot \hat{f}_{\tau^t}(x_t, \cdot)
$$

$$
+ \sum_{t=d_{\max}+1}^{T} p_t \cdot \left( \ell(x_t, \cdot) - \hat{f}_t(x_t, \cdot) \right) + \sum_{t=d_{\max}+1}^{T} p_\star(\cdot \mid x_t) \cdot \left( \hat{f}_t(x_t, \cdot) - \ell(x_t, \cdot) \right)
$$

$$
+ \sum_{t=d_{\max}+1}^{T} (p_t - p_\star(\cdot \mid x_t)) \cdot \left( \hat{f}_t(x_t, \cdot) - \hat{f}_{\tau^t}(x_t, \cdot) \right).
$$

We bound each term individually in the following lemmas and claims, and then we combine all the bounds to derive Theorem 4.6.

**Claim B.1.** *With probability* 1*, it holds that*

$$
\sum_{t=1}^{d_{\max}} (p_t - p_\star(\cdot \mid x_t)) \cdot \ell(x_t, \cdot) \le d_{\max}.
$$

*Proof.* This follows immediately by the fact that $\ell(\cdot)$ is bounded in $[0, 1]$. $\qquad\square$

**Lemma B.2** (Restatement of Lemma 4.7)**.** *With probability* 1*, it holds that*

$$
\sum_{t=d_{\max}+1}^{T} (p_t(\cdot) - p_\star(\cdot|x_t)) \cdot \hat{f}_{\tau^t}(x_t, \cdot) \le \frac{KT}{\gamma} - \sum_{t=d_{\max}+1}^{T} \sum_{a \in \mathcal{A}} \frac{p_\star(a|x_t)}{\gamma p_t(a)}.
$$

*Proof.* For $t \in \{d_{\max} + 1, d_{\max} + 2, \ldots, T\}$, let $R_t(p)$ denote the objective of the convex minimization problem in Equation (1), i.e,

$$
R_t(p) = \sum_{a \in \mathcal{A}} p(a) \cdot \hat{f}_{\tau^t}(x_t, a) - \frac{1}{\gamma} \sum_{a \in \mathcal{A}} \log(p(a)).
$$

Hence,

$$
(\nabla R_t(p))_a = \hat{f}_{\tau^t}(x_t, a) - \frac{1}{\gamma p(a)}.
$$

Since $p_\star(\cdot|x_t)$ is a feasible solution and $p_t$ is the optimal solution, by first-order optimality conditions we have

$$
\sum_{a \in \mathcal{A}} p_\star(a|x_t) \left( \hat{f}_{\tau^t}(x_t, a) - \frac{1}{\gamma p_t(a)} \right) - \sum_{a \in \mathcal{A}} p_t(a) \left( \hat{f}_{\tau^t}(x_t, a) - \frac{1}{\gamma p_t(a)} \right) \ge 0,
$$

Thus,

$$
\sum_{a \in \mathcal{A}} (p_\star(a|x_t) - p_t(a)) \hat{f}_{\tau^t}(x_t, a) \ge \sum_{a \in \mathcal{A}} \frac{p_\star(a|x_t)}{\gamma p_t(a)} - \frac{K}{\gamma}.
$$

Which implies that

$$
\sum_{a \in \mathcal{A}} (p_t(a) - p_\star(a|x_t)) \hat{f}_{\tau^t}(x_t, a) \le \frac{K}{\gamma} - \sum_{a \in \mathcal{A}} \frac{p_\star(a|x_t)}{\gamma p_t(a)}.
$$

We conclude that

$$
\sum_{t=d_{\max}+1}^{T} (p_t - p_\star(\cdot|x_t)) \cdot \hat{f}_{\tau^t}(x_t, \cdot) \le \frac{KT}{\gamma} - \sum_{t=d_{\max}+1}^{T} \sum_{a \in \mathcal{A}} \frac{p_\star(a|x_t)}{\gamma p_t(a)}.
$$

$\qquad\square$

**Lemma B.3** (Restatement of Lemma 4.8). *It holds true that*

$$\mathbb{E}\left[\sum_{t=d_{\max}+1}^{T} p_t \cdot \left(\ell(x_t, \cdot) - \hat{f}_t(x_t, \cdot)\right)\right] \leq \frac{KT}{\gamma} + \gamma \mathcal{R}_T(\mathcal{O}_{sq}^{\mathcal{F},\eta}).$$

*Proof.* For this term, we apply the oracle expected regret bound for the non-delayed function approximation. By Assumption 4.2 the following holds.

$$\mathbb{E}\left[\sum_{t=d_{\max}+1}^{T} p_t \cdot \left(\ell(x_t, \cdot) - \hat{f}_t(x_t, \cdot)\right)\right]$$

$$\leq \mathbb{E}\left[\sum_{t=d_{\max}+1}^{T} \sum_{a \in \mathcal{A}} p_t(a)\left(\ell(x_t, a) - \hat{f}_t(x_t, a)\right)\right]$$

$$= \mathbb{E}\left[\sum_{t=d_{\max}+1}^{T} \sum_{a \in \mathcal{A}} \sqrt{\frac{\gamma}{\gamma}} p_t(a)\left(\ell(x_t, a) - \hat{f}_t(x_t, a)\right)\right]$$

$$\leq \mathbb{E}\left[\sum_{t=d_{\max}+1}^{T} \sum_{a \in \mathcal{A}} \frac{p_t(a)}{\gamma}\right] + \gamma \mathbb{E}\left[\sum_{t=d_{\max}+1}^{T} \sum_{a \in \mathcal{A}} p_t(a)\left(\ell(x_t, a) - \hat{f}_t(x_t, a)\right)^2\right] \quad \text{(AM-GM)}$$

$$= \frac{(T - d_{\max})K}{\gamma} + \gamma \sum_{t=d_{\max}+1}^{T} \mathbb{E}\left[\left(\hat{f}_t(x_t, a_t) - \ell(x_t, a_t)\right)^2\right]$$

$$\leq \frac{(T - d_{\max})K}{\gamma} + \gamma \sum_{t=1}^{T} \mathbb{E}\left[\left(\hat{f}_t(x_t, a_t) - \ell(x_t, a_t)\right)^2\right]$$

$$\leq \frac{KT}{\gamma} + \gamma \mathcal{R}_T(\mathcal{O}_{sq}^{\mathcal{F},\eta}).,$$

where in the final transition we used Assumption 4.5 which implies that the observations are given to the oracle in the same order that they arrive to the CMAB algorithm, which allows us to invoke the regret guarantee of the non-delayed oracle. $\square$

**Lemma B.4** (Restatement of Lemma 4.9). *It holds true that*

$$\mathbb{E}\left[\sum_{t=d_{\max}+1}^{T} p_\star(\cdot|x_t) \cdot \left(\hat{f}_t(x_t, \cdot) - \ell(x_t, \cdot)\right)\right] \leq \mathbb{E}\left[\sum_{t=d_{\max}+1}^{T} \sum_{a \in \mathcal{A}} \frac{p_\star(a|x_t)}{\gamma p_t(a)}\right] + \gamma \mathcal{R}_T(\mathcal{O}_{sq}^{\mathcal{F},\eta}).$$

*Proof.* For this term, we would like to use a change-of-measure technique using AM-GM to be able to apply the oracle's expected regret bound for the non-delayed function approximation. Again, by Assumption 4.2 the following holds.

$$\mathbb{E}\left[\sum_{t=d_{\max}+1}^{T} p_\star(\cdot|x_t) \cdot \left(\hat{f}_t(x_t, \cdot) - \ell(x_t, \cdot)\right)\right]$$

$$\leq \mathbb{E}\left[\sum_{t=d_{\max}+1}^{T} \sum_{a \in \mathcal{A}} p_\star(a|x_t) \cdot \left(\hat{f}_t(x_t, a) - \ell(x_t, a)\right)\right]$$

$$= \mathbb{E}\left[\sum_{t=d_{\max}+1}^{T} \sum_{a \in \mathcal{A}} p_\star(a|x_t) \sqrt{\frac{\gamma p_t(a)}{\gamma p_t(a)}} \cdot \left(\hat{f}_t(x_t, a) - \ell(x_t, a)\right)\right]$$

$$\leq \mathbb{E}\left[\sum_{t=d_{\max}+1}^{T} \sum_{a \in \mathcal{A}} \frac{p_\star^2(a|x_t)}{\gamma p_t(a)}\right] + \gamma \mathbb{E}\left[\sum_{t=d_{\max}+1}^{T} \sum_{a \in \mathcal{A}} p_t(a)\left(\hat{f}_t(x_t, a) - \ell(x_t, a)\right)^2\right] \quad \text{(AM-GM)}$$

$$\leq \mathbb{E}\left[\sum_{t=d_{\max}+1}^{T} \sum_{a \in \mathcal{A}} \frac{p_\star(a|x_t)}{\gamma p_t(a)}\right] + \gamma \sum_{t=d_{\max}+1}^{T} \mathbb{E}\left[\left(\hat{f}_t(x_t, a_t) - \ell(x_t, a_t)\right)^2\right]$$

$$\leq \left[\sum_{t=d_{\max}+1}^{T} \sum_{a \in \mathcal{A}} \frac{p_\star(a|x_t)}{\gamma p_t(a)}\right] + \gamma \sum_{t=1}^{T} \mathbb{E}\left[\left(\hat{f}_t(x_t, a_t) - \ell(x_t, a_t)\right)^2\right]$$

$$\leq \mathbb{E}\left[\sum_{t=d_{\max}+1}^{T} \sum_{a \in \mathcal{A}} \frac{p_\star(a|x_t)}{\gamma p_t(a)}\right] + \gamma \mathcal{R}_T(\mathcal{O}_{\mathrm{sq}}^{\mathcal{F},\eta}),$$

where in the final transition we used Assumption 4.5 as in Lemma 4.8. $\qquad \square$

We now proceed to prove our final lemma.

**Lemma B.5** (Restatement of Lemma 4.10). *Under Assumption 4.4 it holds true that*

$$\mathbb{E}\left[\sum_{t=d_{\max}+1}^{T} (p_t - p_\star(\cdot \mid x_t)) \cdot \left(\hat{f}_t(x_t, \cdot) - \hat{f}_{\tau^t}(x_t, \cdot)\right)\right] \leq 2\sqrt{d_{\max} D \beta}.$$

*Proof.* Using Hölder's inequality and the triangle inequality, we have

$$\sum_{t=d_{\max}+1}^{T} (p_t - p_\star(\cdot \mid x_t)) \cdot \left(\hat{f}_t(x_t, \cdot) - \hat{f}_{\tau^t}(x_t, \cdot)\right)$$

$$\leq \sum_{t=d_{\max}+1}^{T} \|p_t - p_\star(\cdot \mid x_t)\|_1 \cdot \|\hat{f}_t(x_t, \cdot) - \hat{f}_{\tau^t}(x_t, \cdot)\|_\infty$$

$$\leq 2 \sum_{t=d_{\max}+1}^{T} \sum_{i=1}^{d_{\tau^t}} \|\hat{f}_{t-i}(x_t, \cdot) - \hat{f}_{t-(i-1)}(x_t, \cdot)\|_\infty \qquad (\tau^t = t - d_{\tau^t})$$

$$\leq 2 \sum_{t=d_{\max}+1}^{T} \sum_{i=1}^{d_{\tau^t}} \|\hat{f}_{t-i} - \hat{f}_{t-(i-1)}\|_\infty$$

$$\leq 2 \sum_{t=d_{\max}+1}^{T} \sigma_t \|\hat{f}_t - \hat{f}_{t+1}\|_\infty,$$

where $\sigma_t$ is the number of pending observations (that is, which have not yet arrived) as of round $t$. Now, using the Cauchy-Schwarz inequality, the fact that $\mathbb{E}[\sqrt{\cdot}] \leq \sqrt{\mathbb{E}[\cdot]}$ and Assumption 4.4, we finally obtain

$$\mathbb{E}\left[\sum_{t=d_{\max}+1}^{T} (p_t - p_\star(\cdot \mid x_t)) \cdot \left(\hat{f}_t(x_t, \cdot) - \hat{f}_{\tau^t}(x_t, \cdot)\right)\right] \leq 2\sqrt{\left(\sum_{t=d_{\max}+1}^{T} \sigma_t^2\right) \cdot \left(\sum_{t=d_{\max}+1}^{T} \mathbb{E}\left[\|\hat{f}_t - \hat{f}_{t+1}\|_\infty^2\right]\right)}$$

$$\leq 2\sqrt{d_{\max} D \beta},$$

where we used the fact that $\sigma_t \leq d_{\max}$ for all $t$ (since at every round $\sigma_t$ can increase at most by one and an observation can remain pending for at most $d_{\max}$ rounds), and the fact that $\sum_t \sigma_t = D$, which follows since when summing the delays, each delay $d_t$ contributes once to exactly $d_t$ rounds with pending observations, and all pending observations are covered in this manner. $\qquad \square$

We can now prove Theorem 4.6.

**Theorem B.6** (Restatement of Theorem 4.6). *Let* $\gamma = \sqrt{\frac{KT}{\mathcal{R}_T(\mathcal{O}_{sq}^{\mathcal{F},\eta})}}$. *Then the following expected regret bound holds for Algorithm 1.*

$$\mathbb{E}[\mathcal{R}_T] \leq O\left(\sqrt{KT\left(\mathcal{R}_T(\mathcal{O}_{sq}^{\mathcal{F},\eta})\right)} + \sqrt{d_{\max} D \beta}\right).$$

*Proof of Theorem 4.6.* Putting the results of Claim B.1 (taking expectation on both sides) and Lemmas 4.7 to 4.10 all together, the expected regret is bounded as follows.

$$\mathbb{E}[\mathcal{R}_T] \leq d_{\max} + 2\frac{KT}{\gamma} + 2\gamma\mathcal{R}_T(\mathcal{O}_{\mathrm{sq}}^{\mathcal{F},\eta}) + 2\sqrt{d_{\max}D\beta}.$$

Choosing $\gamma = \sqrt{\frac{KT}{\mathcal{R}_T(\mathcal{O}_{\mathrm{sq}}^{\mathcal{F},\eta})}}$ yields the desired bound. $\qquad\square$

## B.2 Regret and Stability analysis of Vovk's aggregating forecaster for the square-loss

We consider a hedge-based version of Vovk's aggregating forecaster [39], presented in Algorithm 2 for the square loss under the realizability assumption (Assumption 4.1) and a finite function class $\mathcal{F}$.

We denote by $z_t = (x_t, a_t) \in \mathcal{X} \times \mathcal{A}$ the input of each function $f \in \mathcal{F} \subseteq \mathcal{X} \times \mathcal{A} \to [0,1]$ at time step $t \in [T]$, where $x_1, \ldots, x_T \in \mathcal{X}$ is a sequence of contexts generated throughout, and $a_1, \ldots, a_T \in \mathcal{A}$ is the sequence of actions, where $a_i$ was chosen for the context $x_i$, for all $i \in [T]$. Also, let $y_1, \ldots, y_T \in [0,1]$ are such that $\mathbb{E}[y_t|z_t] = f_\star(z_t)$, and $f(z_t) \in [0,1]$ for all $f \in \mathcal{F}$. We consider the square loss, and prove the following guarantee for the iterates of Algorithm 2.

**Theorem B.7** (Restatement of Theorem 4.11). *For $t \in [T]$ denote by $q_t \in \Delta(\mathcal{F})$ the probability measure over functions in $\mathcal{F}$ computed by Algorithm 2 at time step t, for the $t-1$-length prefix of the sequence $\{(z_\tau, y_\tau)\}_{\tau=1}^T$.*

*Then, the sequence of measures $\{q_t\}_{t=1}^T$ satisfies the followings for any $\eta \leq 1/18$:*

*1. Expected regret:*

$$\sum_{t=1}^T \mathbb{E}\left[(f_t(z_t) - f_\star(z_t))^2\right] \leq \frac{2\log|\mathcal{F}|}{\eta}.$$

*2. Stability:*

$$\sum_{t=1}^T \mathbb{E}[KL(q_t\|q_{t+1})] \leq 9\eta^2 \cdot \frac{2\log|\mathcal{F}|}{\eta} = 18\eta\log|\mathcal{F}|.$$

*Proof.* WLOG, since Hedge is invariant under adding a constant loss in each round we can subtract $(f_\star(z_t) - y_t)^2$ from the loss of all functions. In particular, after the subtraction, $f_\star$ has a cumulative loss of 0. Therefore $q_t(f) \propto w_t(f)$, and $w_{t+1}(f) = w_t(f)e^{-\eta((f(z_t)-y_t)^2 - (f_\star(z_t)-y_t)^2)}$. Denote $W_t = \sum_f w_t(f)$.

We have, as $W_1 = |\mathcal{F}|$ and $W_{T+1} \geq 1$ (since $w_t(f_\star) = w_1(f_\star) = 1$ for all $t$), that

$$\log\frac{W_{T+1}}{W_1} \geq -\log|\mathcal{F}|.$$

On the other hand, for small enough $\eta$ (smaller than a constant),

$$\log\frac{W_{T+1}}{W_1} = \sum_{t=1}^T \log\frac{W_{t+1}}{W_t}$$

$$= \sum_{t=1}^T \log\mathbb{E}_{f\sim q_t}\left[e^{-\eta((f(z_t)-y_t)^2 - (f_\star(z_t)-y_t)^2)}\right]$$

$$\leq \sum_{t=1}^T \log\mathbb{E}_{f\sim q_t}\left[(1 - \eta((f(z_t) - y_t)^2 - (f_\star(z_t) - y_t)^2) + \eta^2((f(z_t) - y_t)^2 - (f_\star(z_t) - y_t)^2)^2)\right]$$

$$(e^x \leq 1 + x + x^2 \text{ for } x < 1)$$

$$\leq \sum_{t=1}^T -\eta\mathbb{E}_{f\sim q_t}[(f(z_t) - y_t)^2 - (f_\star(z_t) - y_t)^2] + \eta^2\mathbb{E}_{f\sim q_t}[(f(z_t) - y_t)^2 - (f_\star(z_t) - y_t)^2)^2]$$

$$(\log(1 + x) \leq x)$$

$$= \sum_{t=1}^{T} -\eta \mathbb{E}_{f \sim q_t}[(f(z_t) - f_\star(z_t))^2 + 2(f(z_t) - f_\star(z_t))(f_\star(z_t) - y_t)]$$
$$+ \eta^2 \mathbb{E}_{f \sim q_t}[((f(z_t) - f_\star(z_t))^2 + 2(f(z_t) - f_\star(z_t))(f_\star(z_t) - y_t))^2]$$
$$\leq \sum_{t=1}^{T} -\eta \mathbb{E}_{f \sim q_t}[(f(z_t) - f_\star(z_t))^2 + 2(f(z_t) - f_\star(z_t))(f_\star(z_t) - y_t)]$$
$$+ 9\eta^2 \mathbb{E}_{f \sim q_t}(f(z_t) - f_\star(z_t))^2.$$

Rearranging, we obtain that

$$\sum_{t=1}^{T} \mathbb{E}_{f \sim q_t}[(f(z_t) - f_\star(z_t))^2 + 2(f(z_t) - f_\star(z_t))(f_\star(z_t) - y_t)]$$
$$\leq \frac{\log|\mathcal{F}|}{\eta} + 9\eta \sum_{t=1}^{T} \mathbb{E}_{f \sim q_t}[(f(z_t) - f_\star(z_t))^2].$$

Taking expectation over $y_1, \ldots, y_T$:

$$\mathbb{E}_{y_1, \ldots, y_T}\left[\sum_{t=1}^{T} \mathbb{E}_{f \sim q_t}[(f(z_t) - f_\star(z_t))^2]\right] \leq \frac{\log|\mathcal{F}|}{\eta} + 9\eta \mathbb{E}_{y_1, \ldots, y_T}\left[\sum_{t=1}^{T} \mathbb{E}_{f \sim q_t}[(f(z_t) - f_\star(z_t))^2]\right].$$

If, suppose $\eta \leq 1/18$ then we immediately obtain

$$\mathbb{E}_{y_1, \ldots, y_T}\left[\sum_{t=1}^{T} \mathbb{E}_{f \sim q_t}[(f_t(z_t) - f_\star(z_t))^2]\right] \leq \frac{2\log|\mathcal{F}|}{\eta},$$

and the expected regret of Algorithm 2 can now be bounded by

$$\sum_{t=1}^{T} \mathbb{E}[(f_t(z_t) - f_\star(z_t))^2] = \sum_{t=1}^{T} \mathbb{E}\left[\left(\sum_{f \in \mathcal{F}} q_t(f)(f(z_t) - f_\star(z_t))\right)^2\right]$$
$$\leq \sum_{t=1}^{T} \mathbb{E}\left[\sum_{f \in \mathcal{F}} q_t(f)(f(z_t) - f_\star(z_t))^2\right]$$
$$= \mathbb{E}_{y_1, \ldots, y_T}\left[\sum_{t=1}^{T} \mathbb{E}_{f \sim q_t}[(f_t(z_t) - f_\star(z_t))^2]\right]$$
$$\leq \frac{2\log|\mathcal{F}|}{\eta},$$

where we have used Jensen's inequality. This concludes the proof of part 1. of the theorem.

For the second part, we observe that

$$KL(q_t\|q_{t+1}) = \mathbb{E}_{f \sim q_t}\left[\log \frac{q_t(f)}{q_{t+1}(f)}\right]$$
$$= \eta \mathbb{E}_{f \sim q_t}[(f(z_t) - y_t)^2] + \log \mathbb{E}_{f \sim q_t} e^{-\eta(f(z_t) - y_t)^2}$$
$$= \eta \mathbb{E}_{f \sim q_t}[f(z_t) - y_t)^2 - (f_\star(z_t) - y_t)^2] + \log \mathbb{E}_{f \sim q_t}\left[e^{-\eta((f(z_t) - y_t)^2 - (f_\star(z_t) - y_t)^2)}\right]$$
$$\leq \eta \mathbb{E}_{f \sim q_t}[(f(z_t) - y_t)^2 - (f_\star(z_t) - y_t)^2]$$
$$+ \log \mathbb{E}_{f \sim q_t}[1 - \eta((f(z_t) - y_t)^2 - (f_\star(z_t) - y_t)^2) + \eta^2((f(z_t) - y_t)^2 - (f_\star(z_t) - y_t)^2)^2]$$
$$(e^x \leq 1 + x + x^2 \text{ for } x < 1)$$
$$\leq \eta^2 \mathbb{E}_{f \sim q_t}[(f(z_t) - y_t)^2 - (f_\star(z_t) - y_t)^2)^2] \qquad (\log(1 + x) \leq x)$$
$$= \eta^2 \mathbb{E}_{f \sim q_t}[((f(z_t) - f_\star(z_t))^2 + 2(f(z_t) - f_\star(z_t))(f_\star(z_t) - y_t))^2]$$

$$\leq 9\eta^2 \mathbb{E}_{f \sim q_t}\left[(f(z_t) - f_\star(z_t))^2\right].$$

Therefore, taking expectation over $y_1, \ldots, y_T$ and using part 1. we obtain

$$\sum_{t=1}^{T} \mathbb{E}[KL(q_t \| q_{t+1})] \leq 9\eta^2 \cdot \frac{2\log|\mathcal{F}|}{\eta} = 18\eta\log|\mathcal{F}|,$$

yields the second part of the theorem. $\qquad\square$

## C   Lower Bounds

### C.1   Proof of Theorem 4.3

In this subsection, we present a lower bound indicating that an additional assumption on the oracle is necessary in order to obtain sub-linear regret in the general function approximation setting. The lower bound shows that with no additional assumption on the least squares oracle, any algorithm incurs linear regret in the presence of delayed feedback, even for a constant delay of $d = 1$.

**Theorem C.1** (Restatement of Theorem 4.3). *For any CMAB algorithm* ALG *in the function approximation setting (that is,* ALG *can only access $\mathcal{F}$ via the oracle) there exists a CMAB instance with constant delay $d = 1$ over a realizable loss function class $\mathcal{F}$ with $|\mathcal{F}| = T + 1$ with an online oracle $\mathcal{O}_{sq}^{\mathcal{F}}$ satisfying $\mathcal{R}_T(\mathcal{O}_{sq}^{\mathcal{F}}) = 0$, on which* ALG *attains regret $\mathcal{R}_T = \Omega(T)$.*

*Proof.* Consider a CMAB instance over $\mathcal{X} = \{x_1, x_2, \ldots, x_T\}$, $\mathcal{A} = \{a_1, a_2\}$ and $\mathcal{F} = \{f_1, f_2, \ldots, f_T, f_\star\}$, where $f_\star$ is the true loss function. The functions in $\mathcal{F}$ are sampled randomly as follows:

$$f_\star(x_i, a_1) = \text{Ber}\left(\frac{1}{2}\right), \quad f_\star(x_i, a_2) = 1 - f_\star(x_i, a_1), \quad i \in \{T\},$$

$$f_i(x, a) = \begin{cases} f_\star(x, a), & x = x_i, \\ \text{Ber}\left(\frac{1}{2}\right), & otherwise, \end{cases} \quad i \in [T].$$

The online sequence of contexts is defined by $x_1, x_2, x_3, \ldots, x_T$ in order. We consider an oracle which at round $t$ outputs the function $\hat{f}_t = f_t$. It is easy to see that the least-squares regret of this oracle is zero because $f_t(x_t, \cdot) = f_\star(x_t, \cdot)$. Now, at round $t$, due to the delay, ALG only has relevant information on $x_t$ given by $\{f_1(x_t, \cdot), \ldots, f_{t-1}(x_t, \cdot)\}$, all of which are random i.i.d. $\text{Ber}\left(\frac{1}{2}\right)$ random variables, with the true loss $f_\star(x_t, \cdot)$ being either $(1, 0)$ or $(0, 1)$ with equal probability and independently of the previous observations of ALG. Therefore, however ALG chooses the next action $a^{(t)}$, with probability $\frac{1}{2}$ it will incur a loss of 1 while simultaneously the other action will have a loss of zero. This means that in expectation over the random construction of $\mathcal{F}$, the algorithm will incur $\Omega\left(\frac{T}{2}\right)$ regret. By the probabilistic method, we know that there exists a fixture of $\mathcal{F}$ depending on ALG on which ALG suffers linear regret, as claimed. $\qquad\square$

### C.2   Lower Bounds for Contextual MAB with Delayed Feedback

In this subsection, we establish lower bounds on the expected regret for CMAB with delayed feedback. Our construction is based on the approach of [10] via a reduction from the full information variant with non-delayed feedback using a blocking argument.

We begin with a lower bound for the policy class setting with a finite policy class $\Pi$, which relies on a reduction from the problem of (agnostic) prediction with expert advice, for which known lower bounds exist in the literature (see e.g. [9]).

We then present a lower bound for the realizable function approximation setting with a finite loss function class $\mathcal{F}$, and for that we construct an explicit hard instance for the full-information non-delayed variant, with a regret lower bound of $\Omega\left(\sqrt{T\log|\mathcal{F}|}\right)$.

We remark that while a regret lower bound of $\Omega\left(\sqrt{KT} + \sqrt{D}\right)$ can be immediately inferred from the results of [9] who consider the special case of multi-armed bandits, our goal is to show that the

dependence on $\log |\mathcal{F}|$ where $\mathcal{F}$ appears jointly with the delay dependence. While the dependence of $\sqrt{KT \log |\mathcal{F}|}$ is known to be tight for CMAB with general function approximation, it is nontrivial that the delay dependent term also contains a dependence on $\log |\mathcal{F}|$, which we prove in the construction that follows.

In our construction we consider the full feedback setting, where for each round $t$ and observed context $x_t \in \mathcal{X}$, the learner observes the entire loss vector $(\ell(x_t, a))_{a \in \mathcal{A}}$ after choosing an action $a_t \in \mathcal{A}$.

**Theorem C.2.** *There exists a finite policy class $\Pi \subseteq \mathcal{A}^{\mathcal{X}}$ mapping contexts to actions, a delay sequence $(d_1, \ldots, d_T)$ with maximal delay $d$ and sum of delays $D = \Theta(dT)$ such that for any CMAB algorithm there is instance of the CMAB problem for which the algorithm incurs expected regret*

$$\mathbb{E}[\mathcal{R}_T] \geq \Omega(\sqrt{D \log |\Pi|}).$$

*Proof.* We observe that CMAB with a policy class can be viewed as a special case of the prediction with expert advice framework [9], where each policy corresponds to an expert, provides a prediction for each context. Hence, the classical lower bound of $\Omega(\sqrt{T \log |\Pi|})$ for the full-information expert setting (see Cesa-Bianchi and Lugosi [9], chapter 2) applies in the absence of delays.

Returning to the delayed CMAB problem, construct a delay sequence $(d_1, \ldots, d_T)$ in which $d$ is the maximal delay and $D = \sum_{t=1}^{T} d_t = \Theta(dT)$ as follows:

Divide the time horizon into $T/(d+1)$ blocks, each containing $d+1$ consecutive rounds. For each block $b \in \{0, 1, \ldots, T/(d+1)-1\}$ and each round $\tau \in \{b(d+1), b(d+1)+1, \ldots, (b+1)(d+1)-1\}$, define the delay as $d_\tau = d - (\tau - b(d+1))$. That is, within each block, the delays decrease from $d$ to 0, in this corresponding order. This also implies that $D = \frac{T}{d+1} \sum_{i=0}^{d} i = \frac{T}{d+1} \cdot \frac{(d+1)(d+0)}{2} = \frac{Td}{2}$. This construction ensures that feedback from all rounds within a block is revealed simultaneously at the end of the block.

The loss sequence is constructed as follows: Consider the loss sequence $(\ell_1, \ldots, \ell_{T/(d+1)})$ given by a lower bound construction for prediction with expert advice over $T/(d+1)$ rounds. The loss of the first round of each block $b$ is defined to be $\ell_b$, and remains the same throughout the block. Now, note that given this construction, the algorithm essentially faces a prediction with expert advice problem over $T/(d+1)$ rounds (the rounds on which information is obtained), with loss values in the range $[0, d+1]$. We remark that we can assume without loss of generality that the algorithm fixes a policy $\pi_b$ at the start of block $b$ and uses it to play actions throughout the entire block, as it does not learn new information within the block.

Thus, we can aggregate each block into a single "super-round" of a reduced expert problem. Specifically, for block $b$, define the aggregate loss of each expert $\pi$ as $\ell_b(\pi) = \sum_{\tau=b(d+1)}^{(b+1)(d+1)-1} \ell(x_\tau, \pi(x_\tau))$. Even if we allow the algorithm to observe full feedback, it essentially observes the full aggregated loss vector over actions in each block, so this construction corresponds to a well-defined instance of prediction with expert advice over the $T/(d+1)$ rounds which are the initial rounds of the blocks.

The resulting reduced problem has $T/(d+1)$ rounds with losses in $[0, d+1]$. Applying the lower bound from Cesa-Bianchi and Lugosi [9] to this reduced problem yields:

$$\mathbb{E}[\mathcal{R}_T] \geq \Omega\left((d+1) \cdot \sqrt{\frac{T}{d+1} \log |\Pi|}\right) = \Omega\left(\sqrt{(d+1)T \log |\Pi|}\right) = \Omega\left(\sqrt{D \log |\Pi|}\right),$$

which completes the proof. $\qquad\square$

We now combine this result with the classical lower bound of $\Omega(\sqrt{KT \log |\Pi|})$ for CMAB with bandit feedback and a finite policy class, which is based on reductions from prediction with expert advice (see, e.g., [29, 6, 7, 9]). This yields the following lower bound for CMAB with delayed feedback in the policy class setting:

**Corollary C.3.** *For CMAB with delayed bandit feedback and a finite policy class $\Pi$, the expected regret satisfies*

$$\mathbb{E}[\mathcal{R}_T] \geq \Omega\left(\sqrt{TK \log |\Pi|} + \sqrt{D \log |\Pi|}\right).$$

To prove a corresponding regret lower bound for the realizable function approximation setting, we similarly require a regret lower bound of $\Omega\left(\sqrt{T \log |\mathcal{F}|}\right)$ for the full-information non-delayed variant of the problem. Such a lower bound, however, does not exist in the literature as far as we are aware, so we exhibit an explicit construction in the following lemma.

**Lemma C.4.** *Let $\mathcal{A} = \{a_1, a_2\}$ be action set, and let $\mathcal{X} = \{x_1, \ldots, x_n\}$ be a set of $n$ contexts where $n \leq T$. Then for any CMAB algorithm there exists a finite loss function class $\mathcal{F} \subseteq \{\mathcal{X} \times \mathcal{A} \to [0,1]\}$ of size $|\mathcal{F}| = 2^n$ and a CMAB instance which is realizable with respect to $\mathcal{F}$, on which the expected regret of the CMAB algorithm is lower bounded by*

$$\mathbb{E}[\mathcal{R}_T] \geq \Omega\left(\sqrt{nT}\right) = \Omega\left(\sqrt{T \log |\mathcal{F}|}\right).$$

*Proof.* Across all of the instances which we construct, the context is chosen uniformly at random from $\mathcal{X}$. We define the function class $\mathcal{F}$ as the set of $2^n$ functions $f$ which, for each $x \in \mathcal{X}$, are defined via $f(x, a_i) = \frac{1}{2} - \varepsilon$ and $f(x, a_j) = \frac{1}{2}$ for the other action $a_j \neq a_i$ (that is, each function in $\mathcal{F}$ has a distinct choice of optimal actions across all $n$ contexts), and we choose $\varepsilon = \sqrt{n/100T}$.

Prior to the interaction, a function $f_\star \in \mathcal{F}$ is selected uniformly at random and the losses are defined to be Bernoulli random variables according to $f_\star$, ensuring realizability holds. More specifically, $\ell(x, a)$ will be a Bernoulli random variable with parameter $f_\star(x, a)$ for all $x \in \mathcal{X}, a \in \mathcal{A}$.

Now, by standard arguments of statistical estimation, since the true loss function $f_\star$ was sampled at random, as long as a given context $x \in \mathcal{X}$ has not appeared more than $\Omega(1/\varepsilon^2)$ times, the CMAB algorithm must incur instantaneous regret of $\varepsilon$ conditioned on this context. Since the contexts are sampled uniformly at random and the loss values for one context reveal no information about the loss for different contexts, the algorithm must incur expected regret of at least $\Omega(\varepsilon t)$ on the first $t$ rounds as long as each context has been sampled $o(1/\varepsilon^2)$ times. With high probability, all contexts are sampled sufficiently many times only after $t = \Omega(n/\varepsilon^2)$ rounds, implying that the expected regret of the algorithm over $T$ rounds is lower bounded by

$$\mathbb{E}[\mathcal{R}_T] \geq \Omega\left(\varepsilon \cdot \frac{n}{\varepsilon^2}\right) = \Omega\left(\frac{n}{\varepsilon}\right) = \Omega\left(\sqrt{nT}\right),$$

which concludes the proof. $\qquad\square$

We remark that the construction in the above proof is similar to the lower bound given by [3] for the bandit case, but here the proof is considerably simpler as the algorithm is not required to perform exploration in order to obtain sufficient feedback.

Thus, by combining the lower bound for contextual bandits with function approximation under bandit feedback [3] with the delayed feedback result above using the same reduction as we described in the proof of Theorem C.2, we obtain:

**Corollary C.5.** *For any CMAB algorithm in the realizable function approximation setting over finite loss function classes, there exists a finite function class $\mathcal{F}$ and a distribution over losses which is realizable by $\mathcal{F}$, for which the expected regret satisfies*

$$\mathbb{E}[\mathcal{R}_T] \geq \Omega\left(\sqrt{TK \log |\mathcal{F}|} + \sqrt{D \log |\mathcal{F}|}\right).$$

