# OpenReview forum: "Regret Bounds for Adversarial Contextual Bandits with General Function Approximation and Delayed Feedback"
_NeurIPS.cc/2025/Conference — NeurIPS 2025 spotlight_

### Official Review · Reviewer_pz59 · 2025-06-29

**Clarity:** 4
**Significance:** 3
**Originality:** 3
**Rating:** 5
**Confidence:** 3

**Summary:**

The paper studies a variation of the well-known problem of contextual multi-armed bandits in which each action taken by the agent incurs in a loss that can only be observed with a certain delay $d_t$, possibly adversarially chosen, with respect to the time when the action itself was taken. The work focuses on two specific settings and for each provides algorithms and derives bounds on their expected regret.
The first one (Policy Class Learning_ setting) assumes that the benchmark optimal policy belongs to a known class of cardinality $N$. In this case, the algorithm proposed is an adapted version of EXP4, which incurs in an expected regret of $O(\sqrt{KT\log N} + \sqrt{D\log N})$.
In the second setting (_Online Function Approximation_ setting), instead, the loss function $f_{\star}$ is assumed to belong to a class $\mathcal{F}$, which the agent has access to.  The goal is therefore to compete against the benchmark given by $\pi_{\star}(x)=argmin_{a\in A}f_{\star}(x, a)$. The algorithm relies on an oracle to build an estimate the loss function, which incurs in a regret of $\mathcal{R}_T(\mathcal{O})$, and the total expected regret of the algorithm is $\sqrt{KT\mathcal{R}_T(\mathcal{O})}$. Lastly, they provide an example of an oracle based on Hedge algorithm, further specifying in this case both upper and lower bounds.

**Questions:**

I have the following questions for the authors.

1. On the line of what I wrote in the previous section, I wonder if there are ways to relax the assumption of knowledge of the classes even partially, as for instance assuming $\Pi$ to be infinite.

2. I am curious about the idea of introducing the oracle as part of the DA-FA algorithm. I understand the algorithm still works using other subroutines other than the one detailed in Algorithm 2, and that this phrasing allows the result to be more general, but are you aware of other oracle algorithms possibly achieving the same regret rate or preferable with respect to other criteria?

**Ethical Concerns:**

["NO or VERY MINOR ethics concerns only"]

**Final Justification:**

I think this paper is valid, hence I decided to keep my positive score.

**Limitations:**

The authors have addressed all the limitations.

**Paper Formatting Concerns:**

There are no major formatting issues. I just noticed the following typos:

* There should be some parentheses in the definition of the parameter $\gamma$ in the statement of Theorem 4.6.
* At line 151, I think the constant D should be defined as $D=\sum_{t=1} ^Td_t$.

**Quality:**

3

**Strengths And Weaknesses:**

The paper is technically sound, and it analyses a problem with theoretical relevance, which is part of a broad literature of interest to the community, providing new results and insights on the topic. Furthermore, the setting and the needed assumptions are clearly stated, making it very clear to understand.

The main weaknesses originate from the theoretical assumptions required to derive the bounds, which could be restrictive. Specifically, for both settings, complete knowledge of the classes $\Pi$ and $\mathcal{F}$, respectively, is assumed.

Moreover, if I am not mistaken, there is no formal definition of the quantity $\mathcal{R}_T(\mathcal{O}_{sq}^{\mathcal{F}})$ in the main paper. I think the paper would benefit clarity-wise from a precise definition.

---

> ### Author Rebuttal · Authors · 2025-07-30
>
> We thank the reviewer for their thorough and positive review of our paper. Below we address the reviewer’s main questions and concerns.
>
> **Regarding the full knowledge of $\Pi$ and $\mathcal{F}$**
>
> We would like to clarify that in the policy class framework of contextual bandits, it is standard to assume the underlying policy class $\Pi$ is finite as no structural assumptions are made on the policy class (see e.g. [1,2,3]). In the general function approximation setting, however, nontrivial bounds can be obtained even in cases where the underlying function class $\mathcal{F}$ is infinite, as long as we have access to an online least-squares oracle for the class which is sufficiently stable - that is, our main result in Theorem 4.6 does not require that $\mathcal{F}$ is finite, and in particular the only interface of the CMAB algorithm with the function class is via the oracle $\mathcal{O}_{sq}^{\mathcal{F},\beta}$. As a Corollary from this general theorem (see Corollary 4.13), we consider a particular case in which $\mathcal{F}$ is finite and derive a more explicit bound with dependence on $\log |\mathcal{F}|$. For infinite classes (such as linear functions), the dependence on $\log |\mathcal{F}|$ can often be replaced by other parameters such as the underlying dimension.
>
> **Regarding the definition of the least-squares oracle regret**
>
> The quantity $R_T(\mathcal{O}_{sq}^{\mathcal{F}})$ denotes an upper bound on the expected least-squares regret of the online regression oracle in use, and is defined in lines 206-208 of the main text.
>
> **Regarding the use of an oracle in the DA-FA algorithm**
>
> We emphasize that our main result given in Theorem 4.6 applies in a black-box manner, in the sense that any online oracle can be used in Algorithm 2 and only its regret on the **observed sequence** affects the resulting bound. In particular, even an oracle which does not have a regret guarantee with respect to all sequences, e.g. one which only has a heuristic regret bound or one which works on the realized sequence of observations, would result in a meaningful regret bound for Algorithm 2.
>
> We would like to thank you again for the supportive review of our paper, and would be happy to answer any further questions you may have during the discussion period.
>
> **References:**
>
> [1] Dudik, Miroslav, et al. "Efficient optimal learning for contextual bandits." arXiv preprint arXiv:1106.2369 (2011).
>
> [2] Agarwal, Alekh, et al. "Taming the monster: A fast and simple algorithm for contextual bandits." International conference on machine learning. PMLR, 2014.
>
> [3] Auer, Peter, et al. "The nonstochastic multiarmed bandit problem." SIAM journal on computing 32.1 (2002): 48-77.

---

> > ### Comment · Reviewer_pz59 · 2025-08-04
> >
> > I thank the authors for their attentive response to the points I have raised. I remain of the idea that this is a valid paper that offers significant contributions, and I will keep my current score.

---

### Official Review · Reviewer_dxPp · 2025-06-30

**Clarity:** 3
**Significance:** 3
**Originality:** 3
**Rating:** 5
**Confidence:** 4

**Summary:**

This paper provides a comprehensive analysis of adversarial Contextual Multi-Armed Bandits (CMAB) in scenarios with delayed feedback, tackling both finite policy class and general function approximation frameworks. The authors highlight a key discrepancy between this formulation and its non-delayed counterpart: standard relatively weak assumptions on a regression oracle are insufficient in the presence of delays, introducing instead a stability condition as a necessary criterion for achieving sublinear regret.

Under this stability criterion, they successfully establish an optimal regret bound in the finite policy setting and a near-optimal regret bound in the general function approximation scenario. In the latter case, a lower bound is provided. Specifically, the authors employ a Hedge-based version of Vovk’s aggregating forecaster, demonstrating its satisfaction of the stability condition, and thereby justifying its efficacy as a practical oracle. The aggregating forecaster matches the lower bound up to a factor of $\sqrt{d_{\max}}$ in the delayed term.

**Questions:**

1. Stability over the delay length appears pivotal in the oracle’s analysis. Is the global stability assumption on the oracle (i.e., stability over all intervals of length T) fundamentally necessary for sublinear regret? Could a weaker assumption, such as max inf-norm stability restricted to intervals of length $\leq d_{\max}$ suffice instead?
2. To what would you attribute the slack between the upper and lower bounds on the regret for the aggregating forecaster? Is it related to the comment above? A quirk of the aggregating forecaster in particular? Is it more likely slack in the lower bound, or something else entirely?

**Ethical Concerns:**

["NO or VERY MINOR ethics concerns only"]

**Final Justification:**

As discussed in my comment below. An aspect I previously considered to be a weakness of this work has been convincingly indicated as a fundamental barrier in this problem setting, and all of my questions have been answered satisfactorily. I concur that it should be accepted.

**Limitations:**

Yes.

**Paper Formatting Concerns:**

No concerns.

**Quality:**

3

**Strengths And Weaknesses:**

Strengths:
- The authors have pinpointed a fundamental difference in the requirements of a regression oracle in case of delayed feedback, and demonstrated the sufficiently of a stability condition on the regression oracle to achieve sublunar regret.
- The contributions lay the foundation for the analysis of more complex settings, such as learning in MDPs with function approximation, and
- The paper is well written, and contributions are clearly delineated.
- The structure of the paper is such that ideas are built up gradually, and consequently renders the paper very easy to follow.
- Complementing results with a lower bound has narrowed down the remaining  slack to a relatively small subset of possibilities.

Weaknesses:
- A few of the assumptions are standard, but arguably strong already (e.g. realisability), but this compounded with the FIFO assumption does limit the applicability of results.
- The discussion on the necessity of the FIFO assumption does lack a bit of nuance, and I do think that weakening of this criteria may still lead to a setting where sublinear regret is possible (for example, if the learner has other means to weakly attribute ordering of feedback).
- The main regret bound is still off by a factor of $\sqrt{d_{\max}}$, which isn’t discussed in very great detail. Given the contributions, this is a minor consideration.

---

> ### Author Rebuttal · Authors · 2025-07-30
>
> We thank the reviewer for their thorough and positive review of our paper. Below we address the reviewer’s main questions and concerns.
>
> **Regarding the FIFO assumption**
>
> While we agree that this assumption might be limiting in some cases, it seems highly unlikely to us that this assumption can be dropped completely in general. The intuitive reason lies in the fact that the interface between the CMAB algorithm and the least-squares oracle heavily depends on the order of the observed observations, which makes it difficult to relate the sequence of observations to the undelayed sequence without the FIFO assumption. One possibility would be to work in batches of length $d_{\max}$ and wait for pending observations, but such an approach would result in a regret bound of the form $\sqrt{d_{\max} KT}$ which is undesirable as it contains a multiplicative dependence of the maximal delay and the number of arms which we wish to avoid. The reviewer also suggests that the FIFO assumption may perhaps be weakened to still obtain meaningful results - this seems to us like a great question for future research and we are currently unsure about the extent to which the FIFO assumption can be relaxed.
>
> **We now address the questions raised by the reviewer:**
>
> 1. The reviewer makes an interesting suggestion regarding weakening the $\beta$-stability assumption to require stability over all intervals of length at most $d_{\max}$. Our current approach which seems inherent given our regret decomposition relies on the fact that the sum over $t$ of discrepancies between oracle predictions between rounds $t$ and $t+d_t$ will be suitably bounded. If we only assumed stability across windows of length $d_{\max}$, it would allow the possibility of a drift of order at least $T$ in the oracle’s predictions across the entire horizon, which we are not sure is possible to handle in our current approach.
>
> 2. The gap between the upper and lower bounds in our regret bound seems to us as a result of suboptimality in the upper bound given in Theorem 4.6. That is, we believe that our analysis of the Hedge forecaster is tight, and that using it with perhaps a different CMAB algorithm with improved delay dependence may result in an optimal regret bound without the additional $\sqrt{d_{\max}}$ factor.
>
> We thank you again for the supportive review of our paper, and would be happy to answer any further questions you may have during the discussion period.

---

> > ### Comment · Reviewer_dxPp · 2025-08-05
> >
> > I thank the authors for the informative responses to my comments and questions. In particular, I appreciate the explanation regarding apparent necessity of the FIFO assumption, which I did regard at first pass something of a weakness. Accordingly, I maintain my score.

---

### Official Review · Reviewer_6N2K · 2025-07-02

**Clarity:** 3
**Significance:** 3
**Originality:** 2
**Rating:** 5
**Confidence:** 3

**Summary:**

The paper studies the contextual bandit problem
with delayed feedback; i.e., the algorithm sees the rewards only after some delay
which is determined by an adversary.
In the finite policy case, they show a regret bound of
$O(\sqrt{KT \log|\Pi|} + \sqrt{D \log|\Pi|})$ where $D$ is total delay.
They further show that this bound is optimal.


For the general function approximation setting, with a regression oracle,
they use an approach inspired by the SquareCB algorithm of Foster and Rakhlin to obtain a bound of
$O(\sqrt{KT R_{T}(O)} + \sqrt{d_{\max}D \beta})$
where $R_T(O)$ is the regret of the oracle and $\beta$ is an "oracle stability" parameter
and $d_{\max}$ is the maximum delay.
This result is under the realizability assumption as well as the assumptions that
the delays do not change the order in which feedback is seen (i.e., FIFO).
For finite function classes, they show that $\beta$ can be bounded with
$\log|F|$ using a specific oracle they provide,
obtaining a bound of
$O(\sqrt{KT \log|F|}(O) + \sqrt{d_{\max}D \log|F|})$
For this special case, they also show a nearly matching lower bound which does not
have the $d_{\max}$ term.
In order to implement the oracle, the authors use Vovk's aggregation algorithm and prove
that it satisfies the stability requirements in their results.

**Questions:**

Questions:
- Is there any intuition as to why removing the FIFO assumption may make the problem challenging?
  The explanations in line 257-262 focus on your algorithm rather than the problem statement itself.

**Ethical Concerns:**

["NO or VERY MINOR ethics concerns only"]

**Final Justification:**

The authors answered my questions and I remain supportive.

**Limitations:**

Yes

**Quality:**

3

**Strengths And Weaknesses:**

Strengths:
- The paper studies (in my opinion) an interesting problem.
  Delayed feedback is interesting both from a practical perspective as in many cases
  one cannot observe the consequences of one's actions (e.g., submitting papers to a conference)
  and from a theoretical perspective as it is a simple and nice modification of the original problem.
- The combination of upper and lower bounds for the settings they study
  makes this a solid contribution and the paper is fairly well-written.

Weaknesses:
- I think most of the technical depth of the paper is from prior work and
  perhaps the marginal technical contribution is not necessarily very high.
  In particular, the algorithm itself is heavily based on SquareCB and
  the use of Vovk's aggregating algorithm for the oracle is also based on prior work.

---

> ### Author Rebuttal · Authors · 2025-07-30
>
> We thank the reviewer for their thorough and positive review of our paper. Below we address the reviewer’s main concerns and questions.
>
> **Regarding the technical depth of the paper**
>
> While the technical analysis in our paper builds upon techniques from prior related work, we emphasize that adapting known results in the function approximation setting to the delayed variant is a highly challenging and nontrivial problem. In particular, coming up with a suitable assumption on the online oracle in use which would allow obtaining nontrivial regret bounds is, in our opinion, one of the major contributions of our work. This contribution is highlighted by Theorem 4.3 which shows that an additional assumption on the oracle other than online regret minimization is necessary in order to obtain sublinear regret in the presence of delays, an observation which may seem logical but is highly nontrivial in hindsight. Furthermore, as we show in Lemma 4.12, this stability assumption is satisfied by the Hedge aggregating forecaster with $\beta=\log|\mathcal{F}|$ which is also a nontrivial observation.
>
> **Regarding the FIFO assumption**
>
> While we do not have a formal justification, we highly believe that the FIFO assumption might be necessary in order to obtain nontrivial bounds in the online function approximation setting with delays. The intuitive reason is that in this setting, the algorithm makes use of an online oracle which estimates the losses in a black-box manner, and whose internal workings can be very complicated. The interface between the CMAB algorithm and the online oracle highly depends on the order in which feedback is fed to the oracle, and thus this order can in general affect the oracle’s outputs. If the FIFO assumption is not satisfied, the CMAB algorithm essentially has no choice but to feed the observations into the oracle in an arbitrary order determined by the adversarial delays sequence, or alternatively work in blocks of length $d_{\max}$ each to force a suitable ordering which would result in an undesirable regret bound of the form $\sqrt{d_{\max} KT}$.
>
> We thank you again for the supportive review of our paper, and would be happy to answer any further questions you may have during the discussion period.

---

### Official Review · Reviewer_BoNS · 2025-07-03

**Clarity:** 2
**Significance:** 3
**Originality:** 3
**Rating:** 4
**Confidence:** 2

**Summary:**

This paper studies contextual multi-armed bandits (CMAB) under adversarial delayed feedback. The authors consider two settings: (1) policy class learning with direct access to a finite policy class, and (2) general function approximation via online least-squares regression oracle under a stability assumption. The authors prove matching lower bounds, show that the stability condition is essential, and demonstrate that a simple Hedge-based forecaster satisfies it, positioning their framework as a significant but still partially restrictive step toward practical delayed-feedback bandit learning.

**Questions:**

1. Can you characterize the β-stability parameter for common oracle implementations beyond Hedge? Does ridge regression with time-varying regularization satisfy β-stability?
2. How does your approach compare to the "doubling trick" baseline that simply restarts the algorithm every $d_{\max}$ rounds?*

**Ethical Concerns:**

["NO or VERY MINOR ethics concerns only"]

**Limitations:**

yes.

**Paper Formatting Concerns:**

No.

**Quality:**

3

**Strengths And Weaknesses:**

## Strengths

- **Practically motivated**: Delayed feedback is the norm in applications like ad placement and clinical trials.
- **Tight theory for policy classes**: Upper and lower bounds coincide, fully characterizing delay complexity.
- **Novel technical ideas** : Stability analysis of Hedge introduces a fresh lens for delayed bandits.  Lower-bound proof uses an inventive optimal-transport argument on Hamming space.
- **General framework**: Extends from finite policy classes to general function approximation via oracle access, which is more practical.

---

## Weaknesses
- **Strong assumptions**
  - *β-stability* (Assumption 4.4) excludes many common online regressors.
  - FIFO arrival order (Assumption 4.5) is rarely met in real networks where packets can be reordered..
- **Presentation hurdles**
  - Key intuitions behind the loss estimator and stability lemma are buried in proofs.
  - Relationship between $d_{\epsilon}$ (policy classes) and stability parameter β lacks high-level explanation.
- **Missing empirical evidence & comparisons**
  - No experiments to illustrate constants or runtime.
  - Limited discussion of connections to delayed linear bandits or privacy-aware bandits.

---

> ### Author Rebuttal · Authors · 2025-07-30
>
> We thank the reviewer for their thorough review. Below we address the reviewer’s main questions and concerns.
>
> **Regarding the stability and FIFO assumptions**
>
> While the stability assumption does limit various oracles in standard use, we emphasize that as we prove in Theorem 4.3, some additional assumption on the oracle other than online regret minimization is necessary in the presence of delays. While we do not have a formal lower bound showing the necessity of the FIFO assumption, the interface between the CMAB algorithm and the oracle in the function approximation setting makes it seems unlikely that such an assumption can be avoided completely, as the oracle’s outputs may be highly dependent on the ordering of the observations, which in turn depends on the trajectory of actions performed by the CMAB algorithm.
>
> **Regarding the presentation**
>
> We thank the reviewer for pointing out unclarities in our presentation. We will make sure to provide more intuition on the loss estimators for the policy class setting in addition to the explanation in lines 171-177. We will also make sure to add some high-level intuition about the connection between the stability of the algorithm in the policy class setting and the $\beta$-stability in the function approximation setting.
>
> **Regarding empirical evidence**
>
> We emphasize that this is a purely theoretical paper whose goal is to provide theoretical guarantees for CMAB algorithms in the presence of delayed feedback, which has not been studied in the general function approximation setting prior to this work.
>
> **Regarding additional comparisons**
>
> We will make sure to include additional discussion on delayed feedback in linear bandits. We will also mention existing works on privacy-aware bandits, though this area of research is not as directly relevant to our work as we mainly focus on delayed feedback in the general function approximation setting of CMAB.
>
> **We now address the questions raised by the reviewer:**
>
> 1. The reviewer raises a very interesting question regarding the $\beta$-stability parameter for other oracles other than Hedge. We remark that in the linear function approximation setting, the standard least squares oracle seems to satisfy such a stability assumption, as is implicit in the analysis of the linear case, see e.g. [1] who study delayed feedback in linear bandits. In this setting, the inverse covariance matrices of the data are monotone in time, which allows relating the least squares estimator with respect to the delayed feedback to the up-to-date estimator in a manner which ultimately results in a suitable stability property. We emphasize that refining the stability assumption and analyzing the stability parameters of existing oracles in use seems like a very interesting question for future research.
>
> 2. We assume that by “Doubling-trick” the reviewer refers to the “buckets reduction" in which $d_{\max}$ copies of the non-delayed CMAB algorithm are run in parallel, and each copy is applied in the appropriate block. This approach would yield a regret rate of $O(\sqrt{d_{\max}TK \cdot R(O)})$ in which there is a multiplicative dependence on both $d_{\max}$ and the number of arms $K$. One of our main goals was to avoid such a multiplicative dependence, and to obtain a sum of two bounds: one that depends only on the delays and the other of the form $\sqrt{KT}$ which is the optimal bound without delays.
>
> We thank you again for the informative review of our paper, and would of course be happy to answer any further questions you may have during the discussion period.
>
>
> **References:**
>
> [1] Vernade, Claire, et al. "Linear bandits with stochastic delayed feedback." International Conference on Machine Learning. PMLR, 2020.

---

> > ### Comment · Reviewer_BoNS · 2025-08-04
> >
> > Thank you for your answers.  I will continue supporting the paper's acceptance by rasing score.

---

### Decision · Program_Chairs · 2025-09-17

**Decision:**

Accept (spotlight)

**Comment:**

This paper studies contextual multi-armed bandits (CMAB) under adversarial delayed feedback, i.e., the algorithm sees the rewards only after some delay which is determined by an adversary. The authors provide matching upper- and lower-bounds for the problem in two classical settings (finite policy, and general policy with some assumptions) proving the optimality of their techniques.
All reviewers are in agreement that not only the problem studied is well-motivated and very interesting, but also appreciate the novelty of the techniques of this works and the matching bounds and believe the paper is of importance to the general AI/ML community.  Therefore, we recommend accepting the paper for NeurIPS. Congratulations!
The authors however are encouraged to pay attention to the feedback provided by the reviewers and incorporate it into the final version of the paper.